# Muscle Activity during Passive and Active Movements in Preterm and Full-Term Infants

**DOI:** 10.3390/biology12050724

**Published:** 2023-05-15

**Authors:** Irina Y. Dolinskaya, Irina A. Solopova, Dmitry S. Zhvansky, Damiana Rubeca, Francesca Sylos-Labini, Francesco Lacquaniti, Yury Ivanenko

**Affiliations:** 1Institute for Information Transmission Problems, Russian Academy of Sciences, Moscow 127994, Russia; 2Moscow Institute of Physics and Technology, Dolgoprudny 141701, Russia; 3Laboratory of Neuromotor Physiology, IRCCS Santa Lucia Foundation, 00179 Rome, Italy; 4Department of Systems Medicine and Center of Space Biomedicine, University of Rome Tor Vergata, 00133 Rome, Italy

**Keywords:** early development, passive movements, shortening reaction, stretch response, spontaneous movements, muscle tone, preterm infants

## Abstract

**Simple Summary:**

An essential stage in the maturation of the central nervous system and the self-organization of neural circuits occurs during the first year of life. Sensory feedback resulting from interactive and spontaneous movements is instrumental for developing sensorimotor circuits in early infancy, and shorter gestational periods may have an impact on muscle strength and functionality. We examined developmental changes in muscle responses to passive movements and muscle activities during spontaneous movements in preterm and full-term infants by analyzing polymyographic recordings in the lower and upper limb muscles. The major finding of the current study was that there were age-related changes in the manifestation of muscle responses in both preterm and full-term infants. The first six months were the major observational period for the variations in post-term development of muscle activity in preterm infants.

**Abstract:**

Manifestation of muscle reactions at an early developmental stage may reflect the processes underlying the generation of appropriate muscle tone, which is also an integral part of all movements. In preterm infants, some aspects of muscular development may occur differently than in infants born at term. Here we evaluated early manifestations of muscle tone by measuring muscle responses to passive stretching (StR) and shortening (ShR) in both upper and lower limbs in preterm infants (at the corrected age from 0 weeks to 12 months), and compared them to those reported in our previous study on full-term infants. In a subgroup of participants, we also assessed spontaneous muscle activity during episodes of relatively large limb movements. The results showed very frequent StR and ShR, and also responses in muscles not being primarily stretched/shortened, in both preterm and full-term infants. A reduction of sensorimotor responses to muscle lengthening and shortening with age suggests a reduction in excitability and/or the acquisition of functionally appropriate muscle tone during the first year of life. The alterations of responses during passive and active movements in preterm infants were primarily seen in the early months, perhaps reflecting temporal changes in the excitability of the sensorimotor networks.

## 1. Introduction

The postnatal development of sensorimotor networks during the first year of life is characterized by maturation of the musculoskeletal system [1,2] and sensory feedback [3], and also by the proper formation of muscle tone, which is an integral part of all movement. A progressive development of accompanying postural changes from more flexed to more extended limb postures is associated with learning to stand and walk [4]. Early stages of sensorimotor system development are also characterized by the occurrence of spontaneous activity. Sensory feedback resulting from interactive (e.g., mother–child [5]) and spontaneous movements is instrumental for coordination of activity in developing sensorimotor circuits, which has been shown to be essential for the proper formation of neuronal networks in early infancy [6,7,8,9]. Alterations of activity patterns (e.g., due to early injuries to the developing brain) may lead to long-lasting neuronal deficits [10,11,12,13].

A number of studies have assessed manifestations of muscle tone and muscle activity in early infancy, both in full-term and preterm infants. The assessments of movements in infants were typically made based on kinematics, and demonstrate age-related changes in these patterns [11,14,15,16] as well as in the intra- and inter-limb coordination [17,18,19]. The analysis of muscle activity provides additional insights into maturation of the neuromuscular spinal and supraspinal control [20,21,22,23,24]. It is also worth noting that the contraction of a muscle results in the length changes of antagonist and other muscles due to mechanical coupling, which may evoke the corresponding reactions [25] or irradiation of sensory responses [26,27]. Whether and how muscle reactions to length changes reflect the development of muscle tone and support quantitative relationships between motor activity and clinical outcomes remains poorly understood, in part due to high inter-trial variability and a high degree of antagonist coactivation [28,29,30].

It is common to think of the way muscles react to slow changes in muscle length as signs of muscular tone [31,32,33,34,35,36,37,38]. Such reactions can be observed at rather slow velocities, and muscle activity may persist after the movement has stopped. Therefore, they are often referred to as tonic muscle reactions, or as being related to a redistribution of muscle tone across antagonist muscles [39,40,41,42], as opposed to reflex responses to brief perturbations. Muscular responses in infants to mechanical perturbations (e.g., responses to tendon taps [27]) or nerve stimulation (e.g., cutaneous flexion reflex [43]) are typically associated with ‘reflex’ reactions and have relatively short latencies, while muscular responses to slow changes in muscle length may differ in their functional role and underlying mechanisms. It has also been argued that maintaining a limb posture following movement involves distinct neural circuits [44,45]. Muscle tone can be manifested in both resistive and ‘compliant’ behavior, related to movement as a state is related to an action [46]. Muscle responses to passive stretching (StR) and shortening (ShR) may reflect resistive and compliant behavior. Compliant behavior has an important functional significance. Forster [47] was likely the first to consider the functional role of ShR, viewing it as a muscle-length adaptation reflex. Therefore, analyzing both types of responses (StR and ShR) can provide valuable insights into how muscle tone manifests during typical and non-typical motor development.

In preterm infants, some aspects of muscular development, such as muscle power development, may follow a different time course compared with that of infants born at term [48]. Both muscle properties and power as well as muscle functioning or tonic postures [49] may be affected by shorter gestational periods. Such disturbances may be transient (only noticeable, for instance, during the first few months of life) or persistent [50]. In this study, we addressed the following question: are there age-related differences between preterm and full-term infants in muscle reactions? To this end, we examined early manifestations of muscle tone by measuring muscle responses to passive stretching and shortening in both upper and lower limbs in preterm infants (at the corrected age from 0 weeks to 12 months) and compared them to those in full-term infants. To our knowledge, the characteristics of tonic muscle reactions between preterm and full-term newborns have not been compared. However, we have previously described the muscular responses to passive movements in infants who were born at term [25]. In a subgroup of participants, we also assessed spontaneous muscle activity during episodes of relatively large limb movements (comparable with those evoked during passive movements). The rationale for measuring both muscle responses to passive movements and muscle activities during active movements is that we focused our study on the analysis of movement-related muscle activity, which can be measured in these two important types of movements. We also aimed at discovering potential differences between the behavior of preterm infants and that of full-term infants.

## 2. Materials and Methods

### 2.1. Participants

Participants were 54 healthy full-term infants (23 females and 31 males, from 0.5 to 12 months postnatal age) and 30 preterm infants (16 females and 14 males, from 0 to 12 months correction age at the moment of investigation). Eight of these infants were recorded several times (at different age) and they were assigned to different age groups depending on the age of registration (see below). For full-term infants, inclusion criteria were: Apgar score > 7 at 1 and 5 min, without delivery event or perinatal history, no known neurological or musculoskeletal pathology physical, and gestational age greater than 38 weeks [group mean ± SD: 39.5 ± 0.6 weeks]. The exclusion criteria were: congenital malformations, genetic and metabolic diseases, and ongoing mechanical ventilation therapy. For preterm infants, inclusion criteria were: Apgar score > 7 at 1 and 5 min, clinically stable at measurement, and birth > 25 weeks of gestational age. The exclusion criteria for the study were the following: infants with malformations, chromosome aberrations, malignant disorders, or congenital infections. Overall, the characteristics of the preterm infants were: birth at 25–36 weeks of gestational age (GA) (32 ± 3 weeks) and birth weight 609–3200 g (1918 ± 756 g). Six of them were extremely preterm (25–28 weeks of GA), two were very preterm (29–31 weeks of GA), eight were moderately preterm (32–33 weeks of GA), and fourteen were late preterm (34–36 weeks of GA). Experiments were performed at the Moscow Research Institute of Clinical Pediatrics. The infant’s parent gave informed written consent to participate in the study. The study was carried out in accordance with the Declaration of Helsinki for experiments on humans, and the protocol had been approved by the Ethics Committee of the Moscow Research Institute of Clinical Pediatrics (protocol n.14/18).

### 2.2. Experimental Setup

Two experiments (experimental procedures) were performed in the same day in most participants: (a) investigation of muscle responses to passive joint movements, and (b) recording of spontaneous movements. The whole experimental session lasted ~30 min (including a placement of EMG electrodes). In full-term infants, some recordings and analyses of passive and active movements have been included in the previous reports [25,51]. In particular, the data on the StR and ShR in full-term infants have been previously reported [25]. As for the analysis of spontaneous movements in full-term infants, this was different in our previous study [51], since we did not identify the episodes of limb (endpoint) movements. In the present study, we focused our analysis on movement-related EMG characteristics (not including ‘isometric’ contractions or small movements) and, accordingly, we examined the data of the infants in which we reliably identified the specified episodes of limb movements (see below). The results on preterm infants (both for passive and spontaneous movements) have not been previously published.

#### 2.2.1. Muscle Responses to Passive Joint Movements (PM)

All 54 healthy full-term infants and 30 preterm infants took part in this experimental session. Since one of the aims of this study was to examine the effect of age, the infants were classified in different age groups (0.5–3 mo, 3–6 mo, 6–9 mo and 9–12 mo, Table 1, corrected age for preterm infants), as in the previous studies [25,52]. Eight full-term infants participated in the study several times (one infant 4 times, four infants 3 times, and five infants 2 times). Nine preterm infants also participated in the study several times (two infants 5 times, three infants 4 times, three infants 3 times, and three infants 2 times). The interval between the experimental sessions in these infants was 2–6 months and these infants were included in different groups, respectively, so that we performed 70 registrations total for full-term infants and 51 registrations for preterm infants (Table 1).

The infants were tested in a quiet room. They were positioned supine on a typical hospital couch. All infants were examined while awake and alert. If the infant was quiet, then the parent(s) simply watched the experiment. If an infant was agitated, then a toy or communication with the parents was used to achieve a more relaxed state. Typically, we started recording passive movements when the infant was almost motionless for 5 to 10 s prior to registration—that is, when there were no evident spontaneous movements. The procedure for performing passive movements has been described in our previous study [25]. Briefly, the same experimenter made passive movements with each infant. Flexion/extension movements were performed in the knee, hip, ankle and elbow joints (Figure 1A). Passive movements were recorded in both left and right limbs, and they were periodic (typically 4–7 consecutive cycles of flexion/extension) and relatively slow (Figure 1B,C). We started with movements in the hip or knee joint followed by the ankle and elbow joints. The ranges of angular motion and the limb segment orientations are illustrated for one infant in Figure 1A. For the hip joint, the initial position corresponded to an extended leg and the experimenter moved the whole limb with a total hip angular excursion of ~60–70°. For the knee joint, the experimenter held a thigh leg segment immobile with one hand (the hip angle and initial knee angle were both ~135°) while slowly changing the orientation of the shank with the other hand (the range of knee motion was ~120°). For the ankle joint, the leg was about extended (the hip joint angle was ~165–175° and the knee angle was ~160°) and the ankle joint’s range of motion was ~30–40°. For the elbow joint, the arm was resting on the coach and the elbow joint’s range of motion was ~120–140° (Figure 1A). Movements in which the infant spun, strained a limb (resisted to the experimenter), or attempted to turn or lift the head, were excluded. For each joint, 4–7 passive flexion/extension cycles were typically made, and we were able to successfully record passive movements in nearly all joints (Table 1). Table 1 shows the total number of flexion/extension movements in each analyzed joint.

#### 2.2.2. Spontaneous Movements (SM)

In a subgroup of participants, we also assessed spontaneous muscle activity. It was recorded in 28 full-term infants (aged from 2 weeks to 7 months) and 22 preterm infants (25–36 weeks of GA, aged from 0 weeks to 7 months corrected age). Two full-term infants participated in this protocol 2 times and nine preterm infants participated several times (two infants 3 times, seven infants 2 times), so that we performed 30 registrations of SMs total in full-term infants and 33 registrations in preterm infants. DeepLabCut could only successfully track and reconstruct the endpoint motion of a portion of the full-term infants (n = 25) that we had previously recorded (n = 40 in [51]). We have now included the data for 3 infants who were 7 months old, bringing the total to 28 full-term infants. Spontaneous movements are typically observed within 5–7 months after birth [18,51,53], and we classified the infants in age groups similar to those used for analyzing passive joint movements: 0–3 months and 3–7 months (Table 2).

Infants were comfortably placed supine on a standard medical couch. The recordings lasted 3–15 min depending on the behavioral state of the infant (median 7 min). If not stated otherwise, only movements during awake, non-crying state epochs were analyzed.

### 2.3. Data Recording

In both sets of experiments (PM and SM), EMG activity was recorded using the Trigno Wireless EMG System (Delsys Inc., Boston, MA, USA), with an overall gain of 1000, a bandwidth of 20–450 Hz, and a sampling rate 1926 Hz. We recorded bilaterally from the following muscles: biceps brachii (BB), triceps brachii (TB), rectus femoris (RF), biceps femoris (BF), tibialis anterior (TA), and gastrocnemius lateralis (GL). The Trigno bar EMG electrodes were relatively small (5 mm) in order to minimize crosstalk. The Trigno EMG sensors also included the IMU sensors so that we recorded the acceleration signals in the RF and TA sensors (at 148.1 Hz) to characterize slow passive movements in the hip and knee joints [25] (see also below). The skin was cleaned and softly rubbed with alcohol before the electrodes were placed. A digital video camera (Panasonic HC-V760 EE-K, 1920 × 1080 pixels, 50 frames/s) was used to capture all infant movements. The video camera was placed on the side of the infant’s legs at ~2 m from the medical couch and oriented at ~45° relative to the horizon (Figure 1D left panel) in order to capture major limb movements in the corresponding plane (see Section 3). EMG and video recordings were synchronized.

### 2.4. Data Analysis

#### 2.4.1. Muscle Responses to Passive Joint Movements

Detailed information about the definition of flexion/extension movement onsets and durations was previously provided [25]. Briefly, for hip and knee joint movements, we determined the onset of motion of the thigh and shank segments, respectively, using the Trigno IMU acceleration signals (and we double-checked it using video recordings). Since the arm and shank segments (where we recorded the EMG and IMU signals) were essentially motionless during movements of the distal segments (forearm and foot, respectively, in Figure 1A), passive movements for the elbow and ankle joints were analyzed by two experimenters independently (with good correspondence) using video recordings. Phases of flexion and extension were analyzed separately. The EMG signals were rectified and then smoothed by a sliding 20-ms window using the root mean square averaging technique [54]. To characterize muscle responses, we calculated the onset (latency), duration, amplitude, and occurrence of ShR and StR.

The onset (latency) and duration of muscle responses were determined using the following criteria (Figure 1C). First, we started by identifying the fragments of low-level (baseline) activity lasting at least 0.5 s, for which the average EMG amplitude was no more than 1.2 times greater than the minimum level throughout the trial (i.e., was close to or equal to the level of noise in the individual muscle recordings). We then determined the EMG responses using the following algorithm: (1) we looked for the periods of EMG activity exceeding the baseline level at least twice and for at least 30 ms, and (2) if there was less than 50 ms between them, they were combined into a single muscle response. If muscle activity persisted following a change in direction of movement (for example, from flexion to extension), the duration was only calculated from the time it began to the end of this phase. When such EMG activity bursts began during the stretching phase of the corresponding muscle and lasted longer than 100 ms, they were classified as StR. Similar to this, if muscle activity started during the shortening phase and lasted longer than 100 ms, it was classified as ShR. Examples of StR and ShR observed in either one or both antagonist muscles are illustrated in Figure 1C along with the estimated latencies and durations. The latency was defined as the delay between the beginning of an extension or flexion movement and the onset of StR and ShR. The latencies and durations of the StR and ShR were normalized and expressed in percentage of the durations of extension and flexion.

The occurrence of ShR and StR was calculated for each muscle, joint, and infant as the proportion of movements in which these responses were present (relative to the total number of flexion or extension movements, respectively). The data for both the left and right limbs were pooled together. The mean EMG activity (in µV) during the defined interval of the muscle response was used to assess the magnitude of StR and ShR.

#### 2.4.2. Spontaneous Movements

For spontaneous movements, first we defined the episodes of prominent endpoint (wrist and foot) limb movements (movement units, MU) and then we analyzed both MU and EMG activity characteristics of these episodes. For defining the episodes of endpoint movements, we adopted an approach based on a velocity threshold and the amount of motion [18,23].

To compute the endpoint movements, we used DeepLabCut version 2.2 [55] to extract the kinematic data from videos for markerless pose estimation. The software allowed us to train deep neural networks on user-defined labeled data to identify the body parts of interest. We first cropped and trimmed the videos to only encompass the time when the infant moved freely without any limitations or distractions. Periods where the infant turned over or moved because of being distractions from the parents were excluded. These parts of the video were then concatenated prior to applying the neural network. Approximately 25 frames were taken from every video and then selected through k-means clustering for the training dataset. Frames were manually labelled. A residual neural network (resnet50) was then trained on the labels for 310,000 iterations, and the training was stopped when the loss of neural network plateaued in the learning rate. We tracked a total of 19 points, representing parts of head, body, legs, and arms (Figure 1D), and four of these labels (left leg, left arm, right leg, and right arm) were used for the analysis of endpoint movements. The trajectory data has been scaled for the upper and lower limbs to convert the pixel distances to centimeters. We analyzed and labelled 63 videos using median-filtered predictions. To determine the episodes of prominent endpoint movements, the instantaneous 2D velocities were calculated and smoothed with a 0.3 s floating window. The movement episodes (MUs) were then selected according to the following criteria: the endpoint 2D velocity exceeded 0.2 m/s, the episode duration was >0.2 s, episodes were combined into one if the interval between them was less than 0.5 s, and the endpoint excursion during this episode for any of the two coordinates (X, Y) was ≥15% of the body height of the infant. We adopted these specific criteria in order to exclude very small (or noisy) movements from the analysis (velocity threshold and amount of motion are often used to identify the movement units during spontaneous motor activity in infants, e.g., [18,23]). The selected episodes were of relatively large limb motion amplitudes (excursion of 15% of the body height corresponds to more than 8 cm, which is comparable with the amplitude of evoked passive movements). An example of identified MUs is illustrated in Figure 1E. In addition, to verify whether the potential differences in the characteristics of MUs between preterm and full-term infants remain similar if smaller movements are included, we also adopted slightly different criteria for MUs, namely: the endpoint 2D velocity exceeded 0.2 m/s, the episode duration was >0.1 s, episodes were combined into one if the interval between them was <0.3 s, and the endpoint excursion was ≥10% of the body height.

The following kinematic MU characteristics were assessed: MU duration and frequency. These parameters were calculated for each limb and averaged for the left and right sides. In addition, we also assessed whether the MUs were accomplished ‘in isolation’ by each limb (i.e., without a concurrent motion of other limbs) or whether there was a kind of co-activation (temporal overlap) with MUs of different limbs. The MU was considered relatively ‘separated’ if a temporal overlap between this MU with any MU of the other limbs was less than 25% of its duration.

Based on increasing muscle activity correlations (across different muscle pairs) with age in full-term infants reported in our previous study [51], we also compared similar characteristics of EMG activity during movement episodes between full-term and preterm infants: correlation coefficient and coactivation index (CI) of antagonist muscles (RF-BF, TA-GL and BB-TB). The coefficient of linear correlation (r) of EMG activity was analyzed for selected pairs of muscles (RF-BF, TA-GL and BB-TB) during the identified episodes of spontaneous movements (during MUs), and averaged across MUs to illustrate age-related differences (assessed using non-parametric statistics like the Kruskal–Wallis test with Holm-Bonferroni correction). The results for the two sides of the body did not show significant differences, so they were subsequently averaged. The CI was assessed between antagonistic muscle groups using the following formula [56,57,58]:
CI=∑NEMGHj+EMGLj2∗[EMGLjEMGHj]N
where *EMG_H_* and *EMG_L_* represent the antagonist muscle pairs’ highest and lowest activity, respectively, and N is the number of temporal points digitized in the MU. The *CI* was averaged over the entire MU duration (from 1 to *N*) in order to provide a global measure of the coactivity level. When this parameter is used, high *CI* values indicate a high level of activation of both muscles, whereas low *CI* values indicate either a low level activation in both muscles or a high-level activation in one muscle with a low-level activation in the other muscle in the pair [56].

### 2.5. Statistics

Non-parametric statistics were used for data analysis because the experimental data typically did not fit the normal distribution criterion (the Shapiro–Wilk W-test, *p* < 0.05). Descriptive statistics (except for Table 1) included means and 95% confidence intervals of means. To compare independent samples, we used the Kruskal–Wallis test and the Mann–Whitney U test with a Holm–Bonferroni correction. We compared parameters in groups of preterm vs. full-term infants, movements in different joints, during flexion/extension, and in different age groups. To compare two dependent samples, we used the Wilcoxon matched pair test. Using Spearman rank-order correlation, the degree of the relationship between StR and ShR amplitudes and the infants’ age was evaluated. The level of statistical significance was set at 0.05.

## 3. Results

### 3.1. Passive Movements

Infants successfully completed the experimental protocol for most joints of the left and right limbs (Table 1 shows the total number of recorded movements for each group). Induced passive movements were fairly slow (flexion/extension cycle duration ~3–4 s, Figure 1C) and periodic (typically 4–7 consecutive flexion/extension cycles). Figure 1A illustrates an example of passive hip, knee, ankle, and elbow flexion-extension movements. The experimenter did not move the contralateral limb and the infants usually maintained it motionless (although we observed consistent rhythmic responses in the contralateral limb in some infants; see below). Even though the passive joint movements were performed manually, the same experimenter executed movements in all infants in a similar manner. The mean duration of flexion and extension movements was roughly comparable in preterm and full-term infants (~1.5 s in all joints, being slightly shorter for the ankle, probably due to the limited range of angular motion; see Figure 1B).

### 3.2. Muscle Responses to Passive Movements in Preterm and Full-Term Infants

Overall, the results showed that all infants demonstrated EMG responses during imposed passive angular motion, although the responses could vary significantly from cycle to cycle across different joints or in different infants. Furthermore, muscle responses were not restricted to the muscle lengthening reactions (StR), since we often observed prominent and repeatable (from cycle to cycle) EMG activity during muscle shortening as well (ShR) in both preterm and full-term infants. Figure 1C illustrates examples of neuromuscular responses to passive angular movements in the different joints in preterm infants. Below, we first describe the characteristics of muscle reactions in the joints being primarily moved, and we then report the occurrence of consistent muscle reactions in other joints of the ipsilateral and contralateral limbs and their manifestation at different ages (throughout the first year of life).

Muscle responses to passive flexion/extension movements were observed in both full-term and preterm infants beginning from the very early age. They could include mainly ShR or StR in one muscle with no concurrent activity in the antagonist (Figure 1C, left panel) or simultaneous StR and ShR in antagonist muscles (Figure 1C, right panel). Since most of our recorded muscles were bi-articular, ShR in RF was observed during flexion in the hip joint and extension in the knee joint, while ShR in BF was observed during hip extension and knee flexion (Figure 2A). StR was generally more frequently observed than in both preterm and full-term infants than ShR (*p* < 0.01, Wilcoxon T test with multiple testing Holm correction), but ShR was more pronounced in the muscles of the upper limb than the lower limb (*p* < 0.01 Wilcoxon T test, when comparing elbow joint movements to those of the hip, knee and ankle joints separately) (Figure 2A). We compared the occurrence of muscle responses to passive movements in four age groups (0.5–3 mo, 3–6 mo, 6–9 mo, and 9–12 mo). While muscle responses were observed in all age groups (Figure 2A), their occurrence significantly decreased with age in many muscles in both preterm and full-term infants (*p* < 0.03, post hoc Mann-Whitney U test). We did not find significant differences in the occurrence of StR between preterm and full-term infants in all age groups (*p* > 0.4 for all muscles and joints) except for StR in the RF muscle for the 3–6 month age group (Figure 2A). For ShR, it also did not differ for most muscles (*p* > 0.3) except for the thigh muscles (BF and RF) in the 3–6 month age group (where ShR and StR were more prominent in preterm infants, *p* < 0.01, post hoc Mann–Whitney U test, Figure 2A). However, within the group of preterm infants of 3–6 months (n = 19), we did not find significant differences in responses related to the gestational age: 7 infants with GA < 32 weeks and 12 infants with GA of 32–36 weeks showed similar occurrence of muscle responses (ShR and StR in RF and BF leg muscles). The abovementioned differences between preterm and full-term infants (group of 3–6 mo, Figure 2A) were also detected when the percentage of infants who demonstrated ShR or StR in at least one or more movement cycle was plotted (Figure 2B): both for ShR and StR, the proportion of such trials was higher in the preterm group (*p* < 0.001 for ShR and *p* < 0.002 for StR, chi-square test).

Concerning the characteristics of EMG responses (duration, latencies, and amplitude), we did not find significant differences in the duration of ShR or StR between preterm and full-term infants (*p* > 0.3 for all muscles and joints, except for knee flexors, with the Mann–Whitney U test) (Figure 2C). There were also no significant differences in latencies of muscle reactions between preterm and full-term infants (*p* > 0.4 Mann–Whitney U test). As a rule, the latencies for ShR (with respect to the onset of the flexion/extension phase) were somewhat longer than for StR in all joints (*p* < 0.04 for all muscles, Wilcoxon T test). In particular, on average, in full-term and preterm infants, the latencies were ~450 ms and ~640 ms for StR and ShR, respectively (Figure 3A). The age had no significant influence on StR and StR latencies (*p* > 0.1, effect of group, with the Kruskal–Wallis test). The beginning of ShR and StR typically occurred between 10% and 50% of movement duration when normalized to the entire length of the extension or flexion movement (Figure 3B). As for the mean amplitude of EMG responses (in μV), it was on average larger during StR than during ShR in several muscles for both preterm and full-term infants. For instance, for full-term ones, the StR activity was larger for arm muscles (~15 μV for StR vs. ~10 μV for ShR), knee flexors (10 μV vs. 7 μV), and ankle dorsiflexors (20 μV vs. 13 μV) (*p* < 0.02 for all these muscles, Wilcoxon T test). For preterm infants, it was also larger for arm extensors (17 μV vs. 13 μV), knee flexors and extensors (10 μV vs. 7 μV), ankle dorsiflexors (23 μV vs. 15 μV), and ankle plantarflexors (11 μV vs. 8 μV) (*p* < 0.04, Wilcoxon T test). However, even though the amplitude of EMG activity in µV can be considered only as a qualitative parameter due to potential differences in skin impedance across infants, we did not find significant differences in the amount of activity of StR and ShR between preterm and full-term infants (*p* > 0.9, Mann-Whitney U test).

Finally, since irradiation of sensory-evoked responses to distant muscles has been reported in infants [25,59], we also looked for and compared (between preterm and full-term infants) the incidence of consistent muscle responses (across consecutive movement cycles) in the joints not being primarily moved by the experimenter. Figure 4A (left panels) illustrates the occurrence of regular (rhythmic) muscle activity in the other joint of the ipsilateral limb in two infants during passive movements in the hip and ankle joints (top and bottom examples, respectively). For movements in the knee joint, we considered only TA, since GL is the bi-articular muscle and one cannot rule out its lengthening/shortening during knee joint movements. Table 3 shows the occurrence of rhythmic activity in different distant joints of the ipsilateral limb, while pie charts in Figure 4A show the total percentage of trials with the presence of rhythmic activity in other joints of the ipsilateral limb. Note a decrement of muscle responses with age in all joints for both preterm and full-term infants (Table 3). Regarding contralateral influences, it is interesting to note that in younger infants (0–6 months), there was a relatively frequent incidence of regular muscle responses in the contralateral limb (typically during rhythmic hip joint movements, Figure 3B), while such consistent contralateral responses to passive joint movements were never recorded in older infants. In sum, the manifestation of evoked rhythmic activity in distant ipsilateral and contralateral limb joints was significantly less frequent in older preterm infants (6–12 months corrected age) than in younger infants, as was the case in full-term infants (Figure 4, Table 3).

### 3.3. Selected Movement Episodes (MUs) of Spontaneous Activity

Spontaneous activity was successfully recorded in 30 full-term and 33 preterm infants (Table 2). Both upper and lower limbs were highly involved in SMs for both groups (see examples of SM recordings in five full-term and five preterm infants in Figure 5A), consistent with previous studies [60,61,62,63]. There were clear limb motions in both the X and Y directions (Figure 5A) that have been used for selecting movement episodes (see Section 2). Movements could be relatively short (<0.5 s) or longer (several seconds) and, during each episode of movement, the infant could change the direction of movement of the limb several times (Figure 5B,C). These movement episodes (MUs) were typically accompanied by pronounced EMG activity in most muscles (Figure 5B,C).

### 3.4. Characteristics of MUs in Preterm and Full-Term Infants

Despite high variability, the frequency and duration of selected MUs were generally similar for all age groups: the mean frequency was ~6–9 MUs per minute, and mean duration was ~1–1.5 s (Figure 6A). Furthermore, we did not find significant differences when comparing the groups of full-term and preterm infants. Since all limbs were highly involved in SMs (Figure 5A), we also assessed whether MUs were accomplished ‘in isolation’ by each limb or there was a kind of synchronization (temporal overlap) with the MUs of different limbs. Figure 6B (left panel) illustrates an example of left and right leg MUs during a 30-s period of SMs in one infant (3 months). The MU was considered relatively ‘separated’ if a temporal overlap between this MU with any MU of other upper and lower limbs was less than 25% of its duration (see Section 2.4.2). The percentage of ‘separated’ MUs was relatively small (~20–30%, Figure 6B right panel), although the Kruskal–Wallis test did not reveal a significant trend with age in all groups of infants (*p* > 0.1). Comparison between full-term and preterm infants also did not show significant differences (*p* > 0.1 for all age groups, post hoc Mann–Whitney U test). We also verified how the characteristics of MUs would change if different criteria were used, including smaller movements by decreasing the thresholds for their identification (with endpoint excursion ≥10% of the body height, see Section 2.4.2) resulted in the greater overall number of identified MUs. However, the characteristics of MUs (mean frequency ~10–13 MUs/minute, mean duration ~0.8–1.1 s) remained similar between preterm and full-term infants. The percentage of ‘separated’ MUs (~20–30%) also remained not significantly different between the groups.

Movement episodes (MUs) were associated with pronounced EMG activity (Figure 5B,C). Some muscle activity could also be observed in the intervals between MUs (possibly due to some ‘isometric’ contractions or minor movements or because we identified MUs only in the plane of video camera, not in 3D). Nevertheless, it is worth stressing that a great amount of EMG activity was associated with identified MUs (Figure 5B,C). We have previously reported increasing muscle activity correlations with age during the whole period of SMs in full-term infants [51]. Therefore, in this study we also adopted and assessed similar characteristics of EMG activity during MUs: correlation coefficient (r) and coactivation index (CI) of antagonist muscles (RF-BF, TA-GL and BB-TB). Figure 6C illustrates the results of this analysis.

Overall, a comparison between full-term with preterm infants showed no significant difference for any age group (*p* > 0.1, post hoc Mann–Whitney U test). For the arm antagonist muscles (BB-TB pair), the Kruskal–Wallis test did not reveal significant differences (*p* > 0.1) in r between different age groups (0–3 mo, and 3–7 mo) in both full-term and preterm infants (Figure 6C left panel). However, for the lower limb muscles, we found differences. In particular, for full-term infants, there was a significant increase of r for TA-GL (*p* = 0.05, Kruskal–Wallis test) and for RF–BF (r = 0.03) with age, but for preterm infants, we did not find this effect (Figure 6C). Despite the absence of these age-related increases in preterm infants (Figure 6C), we also could not identify any significant correlation between the selected SM characteristics and the gestational age in our group of preterm newborns (Figure 6D). As for the CI (Figure 6C right panels), the Kruskal– Wallis test did not show a significant effect of age for all muscle pairs (*p* > 0.1). Post hoc analysis with the Mann–Whitney U test did not reveal differences between full-term and preterm infants (*p* > 0.1).

## 4. Discussion

We examined developmental changes in muscle responses during passive (Figure 2, Figure 3 and Figure 4) and active (Figure 5 and Figure 6) movements in preterm and full-term infants by analyzing polymyographic recordings in the lower and upper limb muscles. Below, we discuss the results in the context of the general developmental trend in the characteristics of movement-related muscle activity, which may reflect the processes underlying maturation and self-organization of neural circuits in this period of life.

### 4.1. Limitations of the Study

The electromyographic (surface EMG) technique that we used to evaluate muscle responses has some limitations. For passive movements, we used manual flexion and extension joint movements, since these manipulations are frequently used to assess spasticity in a clinical setting [64,65]. The same experimenter performed measurements in all infants in a similar manner, so that movements were relatively slow and with similar durations in all joints and subject groups, i.e., about 1.5–2 s (Figure 1B. Even though we used non-normalized EMG data, both StR and ShR were determined relative to the individual baseline level of activity in each muscle (see Section 2). Even though some amount of crosstalk can be present in the surface EMG recordings, the correlations of EMG activities of antagonist muscles were rather low on average (e.g., see Figure 6C), and we often observed separate StR and ShR bursts in the antagonist muscles (Figure 1C and Figure 4B). It is also worth noting that crosstalk could not account for frequent rhythmic responses in distant muscles of the ipsilateral (Figure 4A) and contralateral limbs (Figure 4B).

For spontaneous movements, we used 2D video recordings to identify the episodes of SM, which could decrease to some extent the number of identified MUs. Nevertheless, the same procedure was applied in both preterm and full-term infants to select movement episodes, in such a way that they reflected only pronounced limb endpoint motion (see Section 2.4.2).

### 4.2. Comparison of Muscle Activities during Passive and Active (Spontaneous) Movements

We focused our study on manifestations of movement-related muscle activity in early infancy. To this end, we examined the muscle responses to passive joint movements of the upper and lower limbs and the spontaneous movements associated with noticeable upper and lower limb endpoint motions. Muscle activity during these two types of movements (PM and SM) might likely share some characteristics. For example, coactivation of antagonist muscles frequently occurs during passive (Figure 1C) and spontaneous (Figure 6C) movements. Moreover, activity in one limb may evoke or be accompanied by activity in the contralateral limb (Figure 4B and Figure 6B). Both types of muscle responses are also rather prominent in infants during the first few months after birth (Figure 2A and Figure 6A). Nevertheless, the underlying physiological mechanisms are different. Since the infants were examined while awake and alert, it is first crucial to distinguish between muscle responses to passive movements and EMG signals during spontaneous movements.

Indeed, some interference of StR and ShR with ongoing spontaneous activity cannot be ruled out. Nevertheless, it is unlikely that the muscle activity recorded during passive movements was random, i.e., unrelated to the passive movement, for the following reasons. First, in order to achieve the most relaxed state of the infant, typically we started recording passive movements when the infant was almost motionless (for ~5 to 10 s prior to registration), that is, when there were no evident spontaneous movements. Second, even though the experimenter determined the frequency (number) of passive movements (with flexion/extension movement duration ~1.5–2 s, Figure 1B), the occurrence of StR and ShR within the total interval of passive movements recording was significantly higher than the occurrence of spontaneous movements. In particular, the occurrence of the episodes of SMs was about 6–8 MUs/min (Figure 6A left panel), while the occurrence of StR and ShR was much more frequent, approximately every one or two movement cycles (see percent of movements, Figure 2A), which roughly corresponds to ~20–40 muscle responses per minute. Third, and most importantly, these reactions to passive movements were linked to a particular phase of flexion/extension motion, and persisted for several cycles (Figure 1C and Figure 3A,B), meaning that the onset of StR and ShR did not occur randomly throughout flexion/extension motion (Figure 4B) but had a latency of ~0.5 s (Figure 4A). Finally, cyclic flexion/extension movements in one joint could also evoke regular rhythmic muscle responses in distant joints in younger infants (Figure 4). Therefore, we believe that the reported muscle reactions to passive joint motions accurately reflect how muscles respond to gradual changes in muscle length as indicators of muscular tone.

### 4.3. Muscle Responses to Passive Movements

The major finding of the present study was that there were age-related changes in the manifestation of muscle responses to passive movements in both full-term and preterm infants (Figure 2A and Figure 4A,B). Our data support other age-dependent changes in excitability of spinal or supraspinal circuits, such as a reduction in the monosynaptic H-reflex responses [66,67] or in the incidence of responses to mechanical stimulation [43] with increasing age. Interestingly, in younger infants, flexion/extension movements in one joint could also cause rhythmic muscle responses in other ipsilateral or bilateral muscles (Figure 4). This may be influenced by the excitability of the pattern generator circuitry, given the significant functional reorganization of the spinal locomotor output in early infancy [23,30] and the fact that its higher responsiveness is associated with greater H-reflexes [68], which are enhanced in younger infants [66,67]. The so-called “irradiation” of responses to mechanical stimulation to distant muscles in infants [26,27] may also contribute to the appearance of rhythmic responses in distant muscles (Figure 4).

Whatever the exact neural pathways involved in modulating muscle responses to passive movements in infants, it is worth noting their high occurrence. In some muscles, such as BB, TB and BF, they could be observed during every flexion or extension movement, especially in younger age groups (Figure 2A). However, the majority of muscles generally showed a significant decline in the expression of reactions with age in both full-term and preterm infants. We found a relatively higher appearance of responses in the proximal muscles in preterm infants at earlier stages (e.g., at the corrected age of 3–6 months, Figure 2A), possibly due to some ‘temporally’ higher sensitivity of the sensorimotor circuitry. When all infants’ passive movements were combined (Figure 2A), or when the percentage of infants who had ShR or StR in at least one or more cycles of passive movements was plotted (Figure 2B), the difference between preterm and full-term infants (group 3–6 months) could be noticed. Nevertheless, the frequent occurrence of StR and ShR was evident in both preterm and full-term infants (Figure 2, Figure 3 and Figure 4), and the differences almost vanished during 6–12 months of life, since both ShR and StR decreased towards 1 year of life. A relatively low presence of ShR and StR in older infants (Figure 2A) corroborates their relatively low appearance in healthy adults during rest. Considering the expression of muscle responses to slow changes in muscle length as manifestations of muscle tone [31,32,39,69], the high occurrence of StR and ShR in infants during the first several months of life is rather remarkable (Figure 2 and Figure 4).

### 4.4. StR vs. ShR

The way muscles respond to slow changes in muscle length, StR or ShR, is frequently addressed as expressions of muscle tone and its redistribution among antagonists [39,40,41,42]. While the responses to lengthening are most likely associated with the excitability of II afferents from the muscle spindles, the neural substrates of the ShR are not fully understood. The ShR can be induced at relatively low velocities of imposed joint rotations, and the role of Golgi tendon organs [70], group II muscle afferents [71], and joint receptors [34] in its generation have been previously discussed in the context of its functional role as an adaptation reflex of muscle to its length [47]. It has also been noted that ShR is accompanied by a discharge of primary endings of muscle spindles due to α-γ coactivation [72]. Since ShR already belongs to an innate repertoire of compliant motor behavior (observed in infants as young as 0–3 mo, Figure 2A), when cortical control is still immature and limited [73,74], it is reasonable to suggest that it mainly depends on subcortical mechanisms and sensory inputs.

Although the StR in leg muscles tended to occur more often than the ShR (Figure 2A), either separately or within the same extension or flexion movement, nevertheless, both reactions were frequently recorded. Thus, not only resistive (StR) but also ‘compliant’ (ShR) tonic motor reactions are frequently present in infants, suggesting their important role in adaptive motor behavior already during early infancy. Coexistence of prominent StR and ShR should not be considered merely as co-activation, since they could also be observed separately. Coactivation of agonist and antagonist motor units is not a prerogative of motor behavior only in infants and has an important functional significance in adults [36,75]. The axial and limb tone during different postures are sensitively and dynamically modulated, originating from interactions between tonic lengthening and shortening reactions, as well as the mechanical properties of stretched and shortening muscles. As has been stated by Sherrington [76] more than a century ago, “posture follows movement like a shadow”.

The reason why the circuits are more excitable early on may be related to higher excitability of spinal or supraspinal circuits [3,66,67] and learning the functionally appropriate muscle tone (such as maintaining the stationary limb and body postures, developing a dynamic postural frame, and antigravity posture control) during the first months of life, which represents an important phase of maturation of the central nervous system and the processing of sensory information. Variability in the occurrence of ShR vs. StR and their characteristics (Figure 2 and Figure 3) may also be functional and serve to set stable feedback gains [27] and to develop adaptive motor behavior [77] in early infancy.

### 4.5. Muscle Activities during Spontaneous Movements

For active movements, the appearance of selected episodes of relatively large limb movements were in general similar in full-term and preterm infants (Figure 6A,B). However, the age-related increments in correlations of antagonist muscle activity were notable only in full terms (Figure 6C). In our earlier study [51], in which we looked at muscle-muscle correlations over the entire time of SM recording, we found that muscle activity correlations increased with age for full-term infants (mostly for distant muscles). The current findings further demonstrate that this effect is seen in the antagonist muscles during the selected episodes of prominent limb movements (i.e., excluding ‘isometric’ contractions and minor movements). The correlation represents only an ‘integrative’ measure of coordination and, in this study, we analyzed relatively short episodes of spontaneous activity (MUs of ~1.5–2 s duration, Figure 6A). Still, our findings are consistent with some results on the kinematics of SMs [18,78]. For instance, Kanemaru et al. [18] reported a small but significant increment in movement correlation of both arms and legs with age. For preterm infants, the increments in correlation of EMGs of any antagonist group with age were not significant (Figure 6C), probably due to a relatively higher correlation at an earlier age (0–3 mo). However, we could not identify any significant correlation between the selected SM characteristics and the gestational age in our group of preterm newborns (Figure 6D).

Overall, both full-term and preterm infants showed similar characteristics and postterm development of muscle activity during passive and spontaneous movements (Figure 2, Figure 3, Figure 4, Figure 5 and Figure 6). However, the preterm infants showed higher rates of muscle responses at 3–6 months, and lacked age-related increases in antagonist activity correlations during SM (Figure 2A,B and Figure 6C, respectively).

## 5. Conclusions

Overall, post-term development of muscle responses in preterm infants was not much different from that of the full-term infants (although some exceptions were observed, Figure 2A and Figure 6C). Atypical responses in preterm infants were mostly noticeable in the first months, which may have been caused by temporal variations in the excitability of the sensorimotor networks. Further investigations may provide and accumulate additional data on the early development of elements of muscle coordinative behavior by performing longitudinal studies and examining the emergence of the modular muscle control infrastructure [23,30] for detecting signs of early pathology and disturbances in the muscle tone.

## Figures and Tables

**Figure 1 biology-12-00724-f001:**
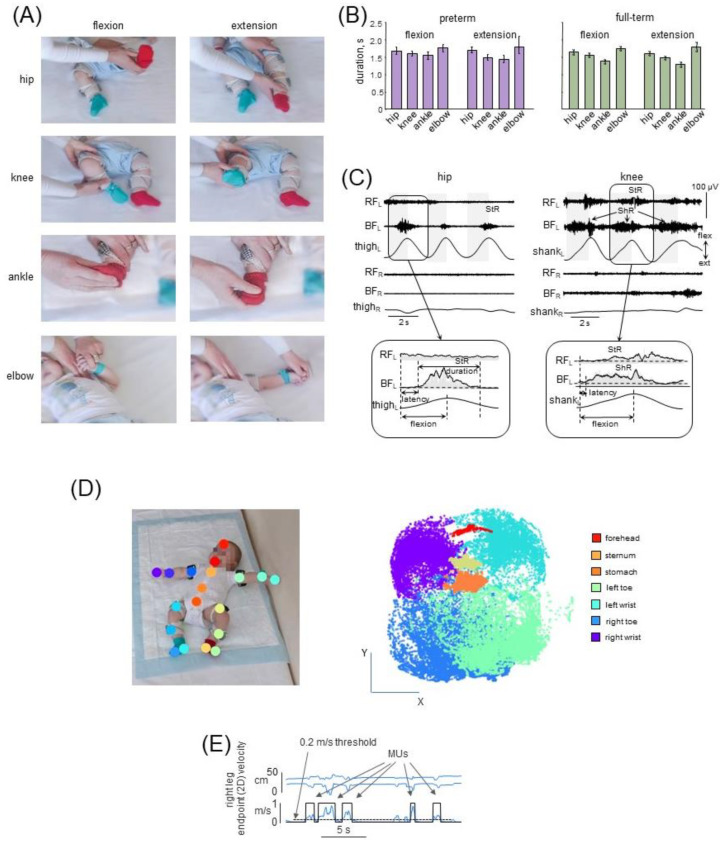
Experimental setup and procedures for analyzing passive (**A**–**C**) and spontaneous (**D**,**E**) movements in preterm and full-term infants. (**A**) Illustration of passive hip, knee, ankle and elbow flexion-extension motions. (**B**) Mean duration (±95% confidence interval across infants) of flexion and extension movements in four joints. The data for the left and right limb motions were pooled together. (**C**) Examples of shortening (ShR) and stretch (StR) responses in the hip and knee joints of two infants, aged 0.5 months and 4.5 months, respectively. The motion of limb segments (thigh and shank) as measured by the IMU (acceleration) is shown. The grey colored areas indicate the duration of the flexion phases. RF—rectus femoris, BF—biceps femoris, R—right, L—left. Lower panels (inserts) refer to the EMG envelopes in one flexion/extension cycle and the calculation of StR and ShR durations (when EMG activity exceeds a threshold denoted by the dotted line, see Section 2.4.1). Note different manifestations of responses: the presence of StR and ShR in either just one muscle (left) or in both antagonist muscles (right). (**D**) For spontaneous movements, we tracked the positions of nineteen markers corresponding to different parts of the body as shown in the left panel. Right panel illustrates an example of selected tracked markers in one infant (3.5 months) during the whole period of recordings (7 min). Four markers (left and right wrists, left and right toes) were used for identifying the movement units (MU) of the upper and lower limbs. (**E**) Example of identified MUs (for the right leg, using toe marker). We selected MUs in which the endpoint 2D velocity exceeded a threshold 0.2 m/s and with excursion >0.15% of the body height (see Section 2.4.2).

**Figure 2 biology-12-00724-f002:**
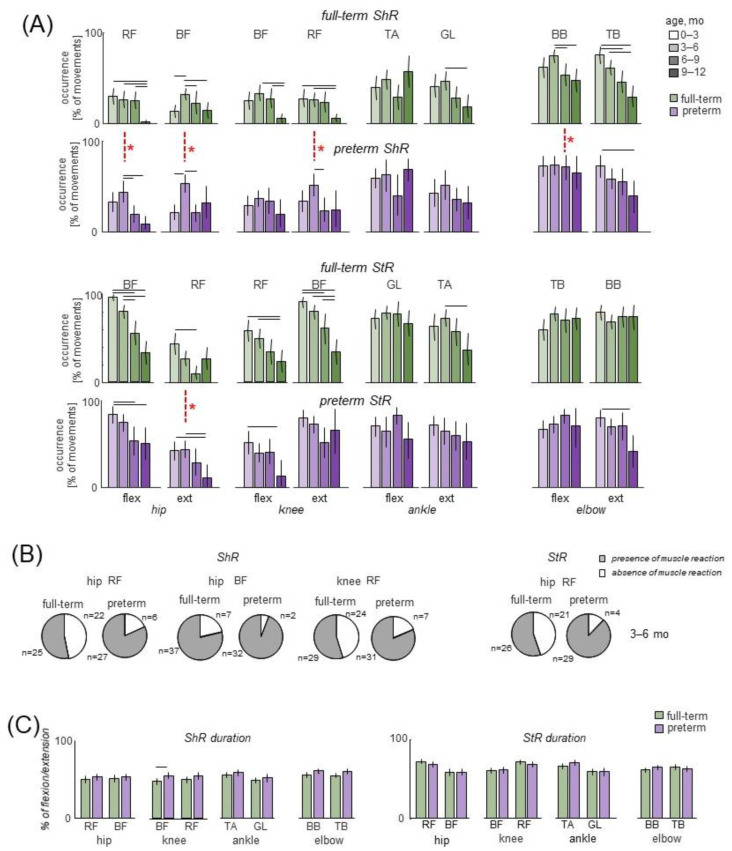
Characteristics of muscle responses to passive movements in different joints in preterm and full-term infants. (**A**) Occurrence of ShR and StR (as the percentage of total number of extension or flexion movements, mean ± 95% confidence interval) during hip, knee, ankle and elbow movements in different age groups (0–3 months, 3–6 months, 6–9 months and 9–12 months). The data for the left and right limb movements were pooled together. Horizontal lines denote significant age differences within each group, while vertical red dashed lines with asterisks denote significant differences between preterm and full-term infants for specific age groups (post hoc Mann–Whitney U test). RF—rectus femoris, BF—biceps femoris, TA—tibialis anterior, LG—gastrocnemius lateralis, BB—biceps brachii, TB—triceps brachii. (**B**) For the age group of 3–6 months that demonstrated significant alterations between full-term and preterm infants (horizontal lines in panel (**A**)), pie charts show the percentage of trials, and their number is also indicated (the trials for the left and right leg joints were pooled together, i.e., summed up), with the presence of ShR or StR in the corresponding muscles in at least one or more cycles of passive movements. (**C**) StR and ShR burst durations (expressed in percent of extension or flexion movement duration, respectively) during passive movements in different joints (since there were no systematic differences with age, the data for different age groups were pooled together).

**Figure 3 biology-12-00724-f003:**
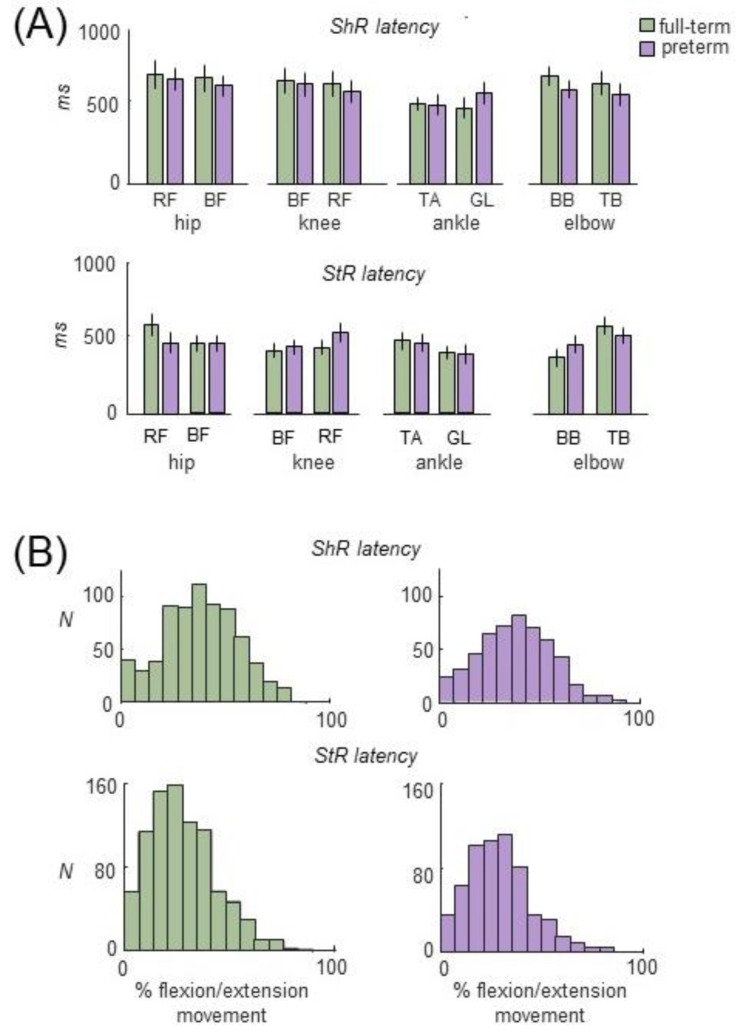
Latencies of ShR and StR. (**A**)—latencies (mean ± 95% confidence interval) of muscle responses in preterm and full-term infants during passive movements in the hip, knee, ankle and elbow joints (the data for different age groups were pooled together as in Figure 2C). (**B**) Histogram of ShR (upper panels) and StR (lower panels) latencies across all infants and all flexion/extension movements. In panel (**A**), latencies were expressed in ms, while in panel (**B**) they were normalized and displayed (*X* axis) in percent of extension or flexion movement duration (to emphasize their most common occurrence between 10% and 50% of movement duration).

**Figure 4 biology-12-00724-f004:**
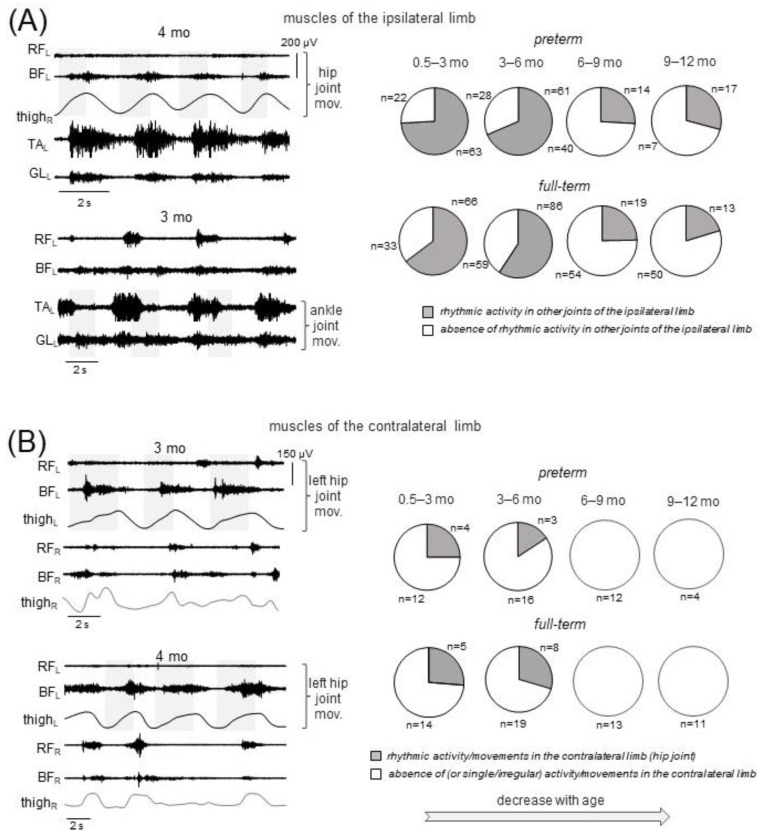
Occurrence of muscle responses in the lower limb joints not being passively moved. (**A**) Left panel: examples of rhythmic-like EMG activity in the ipsilateral limb during flexion/extension movements in the hip (top panel) and ankle (bottom panel) joints in two preterm infants (same format as in Figure 1C). Right panel: pie charts showing the percentage of trials with the presence of rhythmic activity in other joints of the ipsilateral limb for four age groups (for each joint, see the data in Table 3). (**B**) Left panel: examples of rhythmic-like EMG activity and movements in the contralateral limb during left hip joint movements in two preterm infants. Right panel: pie charts showing the percentage of infants with the presence of rhythmic activity in the contralateral limb. Rhythmic muscle responses in the contralateral limb were mainly observed during passive movements in the hip joint.

**Figure 5 biology-12-00724-f005:**
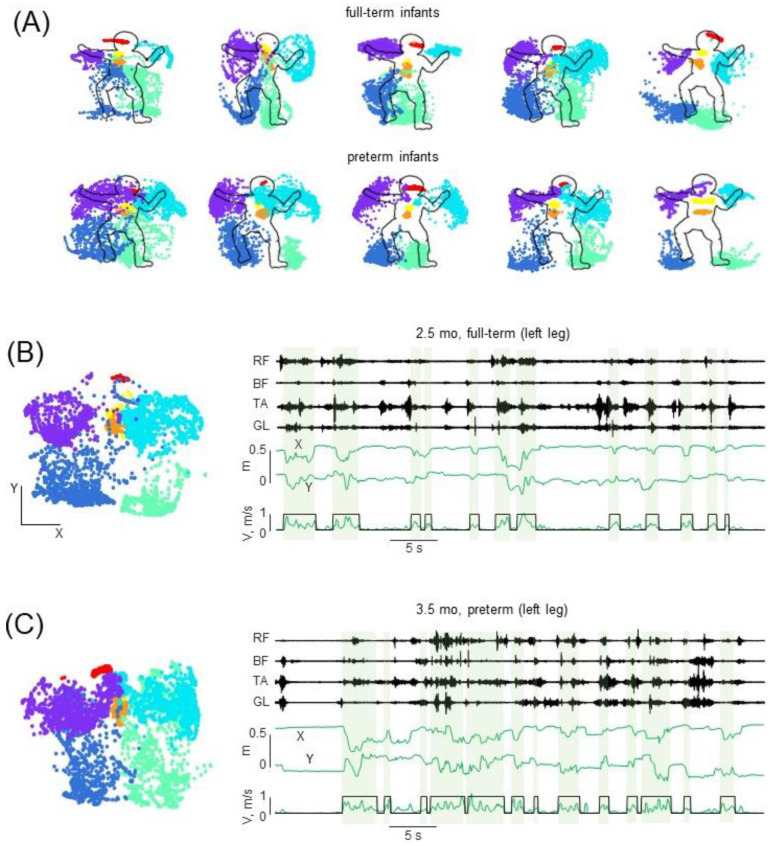
Examples of spontaneous movements. (**A**)—Examples of movements of tracked markers during spontaneous movements (for illustrations, we selected 1-min duration tracking) in five full-term (aged 0.5, 1.5, 1.5, 3, and 5 months, from left to right) and five preterm infants (0.5, 1.5, 2.5, 3.5 and 4.5 months) (same format as in Figure 1D right panel). (**B**,**C**) Examples of 1-min registration of left leg muscle activity and corresponding displacements and velocity of the leg endpoint (toe) in one full-term infant (2.5 months) and one preterm infant (3.5 months), respectively. The identified movement episodes (MUs) are indicated by black curves on the bottom (as in Figure 1E) and also highlighted by shaded green areas. RF—rectus femoris, BF—biceps femoris, TA—tibialis anterior, LG—gastrocnemius lateralis.

**Figure 6 biology-12-00724-f006:**
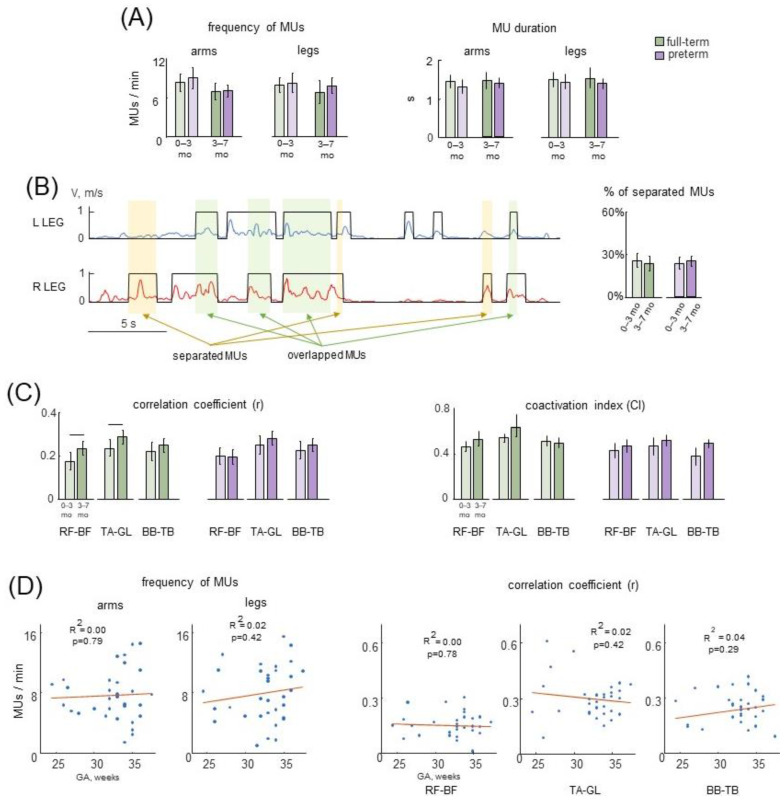
Characteristics of MUs and corresponding EMG activity during spontaneous movements in preterm and full-term infants. (**A**)—mean (±95% confidence interval) frequency (left) and duration (right) of MUs of arm and leg movements for different age groups (0–3 months, and 3–7 months, the data for the right and left limb movements were pooled together). (**B**) Left panel: illustration of separated and overlapped MUs in two legs during a 30-s period of SMs (orange and green shaded areas refer to separated and overlapped MUs, respectively). The right leg was chosen as a referent leg. Right panel: occurrence of separated MUs (as a percent of the total number of MUs) averaged over all limbs and subjects. (**C**) Correlation coefficient (r) and coactivation index (CI) between EMG activities for antagonist muscles (RF-BF, TA-GL, and BB-TB) during selected episodes of SMs (MUs). Horizontal lines denote significant differences. (**D**) Correlations between the selected SM parameters (frequency of MUs, correlation of antagonist muscle activities) and the gestational age (GA) of preterm infants. Each point represents the averaged (across MUs) value for the individual infant. Correlation r and *p* values are reported.

**Table 1 biology-12-00724-t001:** The number of infants in each age group and the total number of passive flexion/extension movements (PM) recorded in different joints (the data for the right and left limb joint movements were summed up). Mean ± SD (across infants, the data for the right and left limbs were averaged) values are also indicated in parentheses.

		PM_Group 1 (0.5–3 mo)		PM_Group 2 (3–6 mo)		PM_Group 3 (6–9 mo)		PM_Group 4 (9–12 mo)	
		N Infants	N Movements	N Infants	N Movements	N Infants	N Movements	N Infants	N Movements
Hip flexion	Full-term	19 16	203 (5.5 ± 1.7) 131 (4.2 ± 1.3)	25 18	241 (5.2 ± 1.2) 175 (5.1 ± 1.3)	13 12	116 (4.5 ± 1.1) 109 (4.9 ± 1.9)	11 4	110 (5.1 ± 1.3) 40 (5.0 ± 1.3)
	Preterm								
Hip extension	Full-term	19 16	201 (5.5 ± 1.6) 126 (4.1 ± 1.3)	25 18	243 (5.3 ± 1.1) 186 (5.5 ± 1.5)	13 12	114 (4.5 ± 1.3) 107 (4.9 ± 1.7)	11 4	103 (4.9 ± 1.3) 42 (5.2 ± 1.0)
	Preterm								
Knee flexion	Full-term	19 16	198 (5.0 ± 1.7) 136 (4.3 ± 1.5)	27 19	239 (4.6 ± 1.3) 181 (4.7 ± 1.3)	13 12	100 (4.1 ± 1.2) 95 (4.3 ± 1.5)	11 4	97 (5.0 ± 1.2) 41 (5.1 ± 1.4)
	Preterm								
Knee extension	Full-term	19 16	195 (5.3 ± 1.8) 138 (4.3 ± 1.5)	27 19	240 (4.8 ± 1.2) 190 (5.0 ± 1.6)	13 12	100 (4.1 ± 1.3) 99 (4.5 ± 1.2)	11 4	92 (4.3 ± 1.2) 41 (5.1 ± 1.3)
	Preterm								
Dorsiflexion	Full-term	15 12	121 (3.9 ± 1.8) 95 (3.9 ± 1.4)	22 9	223 (5.3 ± 1.9) 68 (4.0 ± 1.3)	11 7	95 (4.3 ± 1.9) 63 (4.5 ± 1.3)	10 4	89 (4.2 ± 1.3) 39 (4.9 ± 0.8)
	Preterm								
Plantarflexion	Full-term	15 12	126 (4.2 ± 1.6) 105 (4.3 ± 1.4)	22 9	221 (5.2 ± 2.0) 69 (4.1 ± 1.4)	11 7	95 (4.4 ± 2.0) 65 (4.6 ± 1.2)	10 4	91 (4.3 ± 1.2) 43 (5.4 ± 1.2)
	Preterm								
Elbow flexion	Full-term	19 15	164 (4.4 ± 1.3) 128 (4.3 ± 1.7)	27 18	237 (4.6 ± 1.2) 161 (4.7 ± 1.7)	13 10	113 (4.2 ± 1.2) 91 (4.6 ± 1.4)	11 4	103 (4.7 ± 1.2) 44 (5.5 ± 2.4)
	Preterm								
Elbow extension	Full-term	19 15	157 (4.1 ± 1.2) 118 (4.1 ± 1.5)	27 18	216 (4.4 ± 1.3) 151 (4.4 ± 1.7)	13 10	108 (4.1 ± 1.0) 81 (4.3 ± 1.3)	11 4	95 (4.3 ± 1.2) 44 (5.5 ± 2.3)
	Preterm								

**Table 2 biology-12-00724-t002:** The number of preterm and full-term infants in different age groups (corrected age for preterm infants) recorded during spontaneous movements (SM).

		SM_Group 1 (0–3 mo)	SM_Group 2 (3–7 mo)
N infants	Full-term	17	13
	Preterm	15	18

**Table 3 biology-12-00724-t003:** Occurrence of rhythmic activity in other joints (not being moved by the experimenter) of the ipsilateral leg for preterm and full-term infants in each age group. The number of infants with the presence of activity (and percentage of them in the parenthesis) is indicated.

		PM_Group 1 (0.5–3 mo)	PM_Group 2 (3–6 mo)	PM_Group 3 (6–9 mo)	PM_Group 4 (9–12 mo)
Hip joint movements—activity in TA, GL	Full-term	15/19 (79%)	20/25 (80%)	6/13 (46%)	4/11 (36%)
	Preterm	14/16 (87%)	15/18 (83%)	6/11 (54%)	1/4 (25%)
Knee joint movements—activity in TA	Full-term	14/19 (73%)	19/27 (70%)	5/13 (38%)	5/11 (45%)
	Preterm	14/15 (93%)	13/19 (68%)	6/12 (50%)	3/4 (75%)
Ankle joint movements—activity in RF, BF	Full-term	12/15 (80%)	18/22 (81%)	3/11 (27%)	4/10 (40%)
	Preterm	12/12 (100%)	8/9 (89%)	2/6 (33%)	1/4 (25%)

## Data Availability

Data are available upon reasonable request.

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
