# Peer review of "Muscle Activity during Passive and Active Movements in Preterm and Full-Term Infants"

_biology, 2023, doi:10.3390/biology12050724_

Round 1

Reviewer 1 Report

<A brief summary>

This is an important contribution to our knowledge of sensorimotor development. The authors introduce features and differences in muscle activity that are responsive to passive movements and those that occur during spontaneous movements in full-term and preterm infants. These results suggest that preterm infant have specific developmental course of sensorimotor circuit. These results also suggest that it may be a clinical assessment tool to detect early signs of early pathology and muscle tone disturbances in the early stages of development.

General concept comments

My main concern is that the discussion section does not sufficiently interpret or explain the neurological and physiological mechanisms underlying the results obtained from the experiment. It would be helpful if you could provide arguments and interpretations that compare StR with ShR and combine muscle response to passive movements with muscle activity during spontaneous movements. Additionally, I suggest mentioning at the beginning of the Material and Methods section that the present study partly shares methods and data with your own previous studies. Furthermore, it is important to discuss not only the similarities with previous research, but also the differences observed in the full-term group in this study, including the reasons behind them.

<Specific comments>

[for 1.Introduction at page 2 of 21]

StR and ShR have different mechanisms and functional roles. While StR is a “reflexive” stretch response, ShR is a shortening response, and there are likely to be many differences between them. Please add a description of the neurological significance and mechanism of StR and ShR in or before the paragraph beginning with 'In preterm infants' in the Introduction. Also, analyzing both types of responses can provide valuable insights.

Additionally, could you please explain why you are evaluating spontaneous movements separately from passive movements, and provide a rationale for this? The relationship between muscle responses to passive movements and muscle activities during spontaneous movements is not clear. Also, during spontaneous movements, EMG signals reflect not only muscle responses but also spontaneous/active muscle activities. Therefore, throughout the entire manuscript, it is important to clearly differentiate between muscle responses to passive movements and EMG signals during spontaneous movements.

[Table 1 at page 4 of 21]

The infant group in Table 1 should be almost the same as in the previous study [23], but only Group2 has a few fewer members, and the results are different. It is necessary to explain that this difference is not an arbitrary manipulation.

[2.2.2. Spontaneous movements (SM) at page 4 of 21]

Can you divide the spontaneous movement into monthly groups as cited in reference 31 or add that data to the supplement? Also, please group them by the same age range as the first experiment to allow for comparison with the muscle response to passive movement at the same age.

[for 2.4.1. Muscle responses to passive joint movements at page 5 of 21]

In second paragraph, you defined the threshold of muscle activity. If the threshold of the onset of muscle activity is the mean of the measured interval, the threshold should be higher than the general (frequentlu used) threshold (3 SD at rest) and the sensitivity of the beginning of mscle activity would be lower. Additional explanation is needed why this definition was adopted. Also, please add an explanation what you mean by 1.2 times the minimum value, as the meaning/biological meaning in an experimental system is unclear.

[3.2. Muscle responses to passive movements at page 10 of 21]

You reported that “there were no significantly difference in latencies of muscle reactions between preterm and full-term infants (p>0.4 Mann-Whitney U test).” However, I can not find any data of them. Please provide the data values in the figures or text.

[Figure 4A at page 14 of 21]

It was not clear to me what kind of change you wanted to show here, so please describe what you want to show specifically.

[3.4. Characteristics of MUs and Figure 5 at page 14-16 of 21]

Could you please adjust the range of the group as mentioned earlier? The current range does not allow us to compare with the responses to passive movements, nor to compare it with your own previous study [31]. By modifying the group range, we will be able to make accurate comparisons between these factors.

[4.*. Discussion at page 17-18 of 21]

To make it easier to understand, could you please create subsections for each discussion with titles such as 'Muscle Responses to Passive Movements', 'Muscle Activities During Spontaneous Movements', and 'StR vs ShR'? This will help readers to quickly navigate and understand the different topics being discussed.

[4.*. Discussion at page 17-18 of 21]

It would be interesting for developmental scientists if you could include a subsection titled 'Comparison of Passive Responses with Muscle Activities During Spontaneous Movements'. This would provide valuable insights into the relationship between passive and spontaneous muscle activities, and would further enhance the significance of your findings. As mentioned above, each should have a different neurological background and significance.

[4.*. Discussion at page 17-18 of 21]

Please add the discussion and references about neurological and physiological mechanisms underlying developmental changes of excitability, functional role, and learning during early infancy. At present, there is little prior research or explanation available to help make sense of the developmental changes seen in the current experimental data.

[4.*. Discussion at page 17-18 of 21]

Please add the discussion about the differences of the data in this experiment with the data presented in references [23 and 31], particularly in terms of full-term.

For the discussion about muscle responses to passive movements, although they are the same experiment, there are some differences in occurrences, notably the significant variation at 9-12 months (in Fig 2A vs Fig4 in [23]).

Regarding the discussion on muscle activities during spontaneous movements, I noticed that the age-related increments in correlations between the antagonist muscles of RF-BF and TA-GL were observed across all age groups in your current study, whereas in your previous report they were only observed in infants under 4 months old (Figure 5C vs. Fig2C in [31]). To make the comparison between studies clearer, I suggest dividing the data into monthly groups, as done in reference [31]. This will help readers to better understand the developmental changes in muscle activity across different age groups.

[4.2. Manifestation of muscle responses during the first year of life at page 17 of 21]

You seem to compare the number of spontaneous MU with the passive one determined by the experimenter. Please rewrite it as it was a little confusing.

<Minor points>

I attached the PDF file with my minor comments.

Author Response

This is an important contribution to our knowledge of sensorimotor development. The authors introduce features and differences in muscle activity that are responsive to passive movements and those that occur during spontaneous movements in full-term and preterm infants. These results suggest that preterm infant have specific developmental course of sensorimotor circuit. These results also suggest that it may be a clinical assessment tool to detect early signs of early pathology and muscle tone disturbances in the early stages of development.

We would like to thank the reviewer for an overall assessment of the study, as well as for his/her insightful criticism and suggestions that served to improve the manuscript. We tried to incorporate them all. Changes in the text are marked in red.

<General concept comments>

My main concern is that the discussion section does not sufficiently interpret or explain the neurological and physiological mechanisms underlying the results obtained from the experiment. It would be helpful if you could provide arguments and interpretations that compare StR with ShR and combine muscle response to passive movements with muscle activity during spontaneous movements. Additionally, I suggest mentioning at the beginning of the Material and Methods section that the present study partly shares methods and data with your own previous studies. Furthermore, it is important to discuss not only the similarities with previous research, but also the differences observed in the full-term group in this study, including the reasons behind them.

We now added more discussion and references related to the underlying physiological mechanisms. We also clarified in the Methods which data on fullterm infants have been previously reported and mentioned in the Discussion the similarities. Finally, we reorganized and expanded the discussion section as suggested by the reviewer (see below our responses to specific comments).

<Specific comments>

[for 1.Introduction at page 2 of 21]

StR and ShR have different mechanisms and functional roles. While StR is a “reflexive” stretch response, ShR is a shortening response, and there are likely to be many differences between them. Please add a description of the neurological significance and mechanism of StR and ShR in or before the paragraph beginning with 'In preterm infants' in the Introduction. Also, analyzing both types of responses can provide valuable insights.

As suggested, we added a new paragraph and briefly described the neurological significance of muscle responses to slow changes in muscle length (p2): 

“It is common to think of the way muscles react to slow changes in muscle length as signs of muscular tone [31–38]. Such reactions can be observed at rather slow velocities, and muscle activity may persist after the movement has stopped. Therefore, they are often referred to as tonic muscle reactions, or as being related to a redistribution of muscle tone across antagonist muscles [39–42], as opposed to reflex responses to brief perturbations. Muscular responses in infants to mechanical perturbations (e.g., responses to tendon taps [27]) or nerve stimulation (e.g., cutaneous flexion reflex [43]) are typically associated with ‘reflex’ reactions and have relatively short latencies, while muscular responses to slow changes in muscle length may differ in their functional role and underlying mechanisms. It has also been argued that maintaining a limb posture following movement involves distinct neural circuits [44,45]. Muscle tone can be manifested in both resistive and ‘compliant’ behavior, related to movement as a state is related to an action [46]. Muscle responses to passive stretching (StR) and shortening (ShR) may reflect resistive and compliant behavior. Compliant behavior has an important functional significance. Forster [47] was likely the first who considered the functional role of ShR, viewing it as a muscle-length adaptation reflex. Therefore, analyzing both types of responses (StR and ShR) can provide valuable insights into how muscle tone manifests during typical and non-typical motor development.”

Additionally, could you please explain why you are evaluating spontaneous movements separately from passive movements, and provide a rationale for this? The relationship between muscle responses to passive movements and muscle activities during spontaneous movements is not clear. Also, during spontaneous movements, EMG signals reflect not only muscle responses but also spontaneous/active muscle activities. Therefore, throughout the entire manuscript, it is important to clearly differentiate between muscle responses to passive movements and EMG signals during spontaneous movements.

We focused our study on the analysis of EMG activity since infant movement assessment is rarely based on the EMG signals. We underlined in the Introduction the importance of investigating both ‘tonic’ muscle responses to slow changes in muscle length and muscle activity during spontaneous movements. We now added (at the end of Introduction):

“The rationale for measuring both muscle responses to passive movements and muscle activities during active movements is that we focused our study on the analysis of movement-related muscle activity, which can be measured in these two important types of movements. We also aimed at discovering potential differences between the behavior of preterm infants and that of fullterm infants.”

While these two types of movements have somewhat different mechanisms and cannot be fully differentiated, we provided additional clarifications on differentiating them. In particular, while some interference of StR and ShR with ongoing spontaneous activity cannot be fully ruled out, nevertheless, it is unlikely that they were ‘random’. We discussed this issue in the Discussion. Also, as the reviewer suggested, we now provided new Fig. 3 with information about the latencies, in order to support our view that the onset of StR and ShR was not fully random but started in about 0.5 s after the beginning of flexion or extension motion (Fig. 3). In addition, we often observed rhythmic muscle responses in distant joints (Fig. 4) linked to the specific phase of flexion/extension motion, persistent for several cycles.

[Table 1 at page 4 of 21]

The infant group in Table 1 should be almost the same as in the previous study [23], but only Group2 has a few fewer members, and the results are different. It is necessary to explain that this difference is not an arbitrary manipulation.

We apologize for this inconsistency, in the original submission we indeed slightly decreased the number of fullterm infants in group 2, in part because we recorded significantly more fullterms than preterms in this group (we randomly ‘omitted’ 4 participants and of course we are sure that the results remained similar). However, we agree with the reviewer, therefore now we presented the data of the same infants in group 2 as in our previous study and updated the figures accordingly. Both the results and statistics did not change. In addition, we now used the same format of Fig. 2 as in our previous paper.

[2.2.2. Spontaneous movements (SM) at page 4 of 21]

Can you divide the spontaneous movement into monthly groups as cited in reference 31 or add that data to the supplement? Also, please group them by the same age range as the first experiment to allow for comparison with the muscle response to passive movement at the same age.

As suggested, for SM, we now grouped the infants by similar age range (0-3 and 4-7 mo) as in the first experiment and updated Fig. 5 accordingly. We also divided the infants for the spontaneous movements into monthly groups and provided the results in the supplementary material (however, please consider that we analyzed now less infants for SM than in our previous study [we added this to the Discussion], so that when the decreased total number of participants is divided into six groups, the statistical power is reduced).

[for 2.4.1. Muscle responses to passive joint movements at page 5 of 21]

In second paragraph, you defined the threshold of muscle activity. If the threshold of the onset of muscle activity is the mean of the measured interval, the threshold should be higher than the general (frequentlu used) threshold (3 SD at rest) and the sensitivity of the beginning of mscle activity would be lower. Additional explanation is needed why this definition was adopted. Also, please add an explanation what you mean by 1.2 times the minimum value, as the meaning/biological meaning in an experimental system is unclear.

3SD at rest, as the threshold ‘frequently’ used for the onset of muscle responses, is likely reasonable to use when there is actual background EMG activity prior to the stimulus or movement, while in our case the calculated baseline level was close to or equal to the level of noise in the individual muscle recordings and therefore we adopted a different criterion. (in fact, the threshold we used - 2 times higher than the ‘baseline’ level - was greater than ‘3 SD at rest’)

Likely, our previous description was somewhat incomplete or unclear. Therefore, we now slightly modified this text and replaced “baseline level” with “the fragments of low level (baseline) activity” (p7):

First, we started by identifying the fragments of low-level (baseline) activity lasting at least 0.5 s, for which the average EMG amplitude was no more than 1.2 times greater than the minimum level throughout the trial (i.e., was close to or equal to the level of noise in the individual muscle recordings). Then we …

Also, in our previous study (Solopova et al 2019) we looked at how sensitive the outcomes were to the precise threshold value and we reported that: “We used non-normalized (in µV) EMG data to determine StR and ShR, However, even when we considerably increased the threshold for detecting the periods of muscle activity (three times exceeding the baseline level instead of two times, see Methods), the percentage of detected muscle responses decreased (by 25%), however, the effect of age … remained the same for most muscles.

We believe that our adopted criterion (“2 times higher than the baseline level”) is relevant and preferable to using 3 SD at rest (since the “biological meaning” of ‘3 SD’ around the level of noise might be problematic).

[3.2. Muscle responses to passive movements at page 10 of 21]

You reported that “there were no significantly difference in latencies of muscle reactions between preterm and full-term infants (p>0.4 Mann-Whitney U test).” However, I can not find any data of them. Please provide the data values in the figures or text.

As suggested, we now provided these data (new Fig. 3).

[Figure 4A at page 14 of 21]

It was not clear to me what kind of change you wanted to show here, so please describe what you want to show specifically.

Fig. 4A illustrates some examples of 1-min recordings of SMs, to show that both upper and lower limbs are involved in SMs and that the limb motions in preterm and fullterm infants were notable in both X and Y directions. As suggested, we now mentioned about it in the main text (p.13):  

“Both upper and lower limbs were highly involved in SMs for both groups (see examples of SM recordings in 5 fullterm and 5 preterm infants in Fig. 5A), consistent with previous studies [60–63]. There were clear limb motions in both the X and Y directions (Fig. 5A) that have been used for selecting movement episodes (see Methods).”

[3.4. Characteristics of MUs and Figure 5 at page 14-16 of 21]

Could you please adjust the range of the group as mentioned earlier? The current range does not allow us to compare with the responses to passive movements, nor to compare it with your own previous study [31]. By modifying the group range, we will be able to make accurate comparisons between these factors.

As suggested, we now grouped the infants by similar age range (0-3 and 4-7 mo) as in the first experiment and updated Fig. 5 and the text accordingly (see also our response above).

[4.*. Discussion at page 17-18 of 21]

To make it easier to understand, could you please create subsections for each discussion with titles such as 'Muscle Responses to Passive Movements', 'Muscle Activities During Spontaneous Movements', and 'StR vs ShR'? This will help readers to quickly navigate and understand the different topics being discussed.

We followed the suggestion of the reviewer and added subsections for different topics being discussed.

[4.*. Discussion at page 17-18 of 21]

It would be interesting for developmental scientists if you could include a subsection titled 'Comparison of Passive Responses with Muscle Activities During Spontaneous Movements'. This would provide valuable insights into the relationship between passive and spontaneous muscle activities, and would further enhance the significance of your findings. As mentioned above, each should have a different neurological background and significance.

As suggested, we added this subsection and discussed it (page 17-18).

[4.*. Discussion at page 17-18 of 21]

Please add the discussion and references about neurological and physiological mechanisms underlying developmental changes of excitability, functional role, and learning during early infancy. At present, there is little prior research or explanation available to help make sense of the developmental changes seen in the current experimental data.

We added (p18):

“Our data support other age-dependent changes in excitability of spinal or supraspinal circuits, such as a reduction in the monosynaptic H-reflex responses [66,67] or in the incidence of response to mechanical stimulation [43] with increasing age.”

and p19:

“The reason why the circuits are more excitable early on may be related to higher excitability of spinal or supraspinal circuits ([3,66,67]) and learning the functionally appropriate muscle tone (such as maintaining the stationary limb and body postures, developing a dynamic postural frame, antigravity posture control) during the first months of life, representing an important phase of maturation of the central nervous system and the processing of sensory information. Variability in the occurrence of ShR vs. StR and their characteristics (Figs. 2-3) may also be functional and serve to learn to set stable feedback gains [27] and to develop adaptive motor behavior [77] in early infancy.”

[4.*. Discussion at page 17-18 of 21]

Please add the discussion about the differences of the data in this experiment with the data presented in references [23 and 31], particularly in terms of full-term.

Concerning ‘old’ reference [23], as we responded above, now we presented the data of the same infants in group 2 as in our previous study and updated the figures accordingly. (Both the results and statistics did not change. In addition, we now used the same format of Fig 2 as in our previous paper.)

Concerning ‘old’ reference [31], we added:

“In our earlier study [51], in which we looked at muscle-muscle correlations over the entire time of SM recording, we found that muscle activity correlations increased with age for fullterm infants (mostly for distant muscles). The current findings further demonstrate that this effect is seen in the antagonist muscles during the selected episodes of prominent limb movements (i.e., excluding 'isometric' contractions and minor movements).”

For the discussion about muscle responses to passive movements, although they are the same experiment, there are some differences in occurrences, notably the significant variation at 9-12 months (in Fig 2A vs Fig4 in [23]).

We do apologize for these differences, there was an error in ‘scaling’ for the data for group 4 which indeed overemphasized the effect of age. We corrected this error in new Fig 2A (statistics did not change, we only corrected the bar plots), we now used the same number of fullterm infants as in our previous study (see our response above), and we now adopted the same format of Fig 2A as in our earlier publication to better highlight the effect of age in responses of specific muscles.

Regarding the discussion on muscle activities during spontaneous movements, I noticed that the age-related increments in correlations between the antagonist muscles of RF-BF and TA-GL were observed across all age groups in your current study, whereas in your previous report they were only observed in infants under 4 months old (Figure 5C vs. Fig2C in [31]). To make the comparison between studies clearer, I suggest dividing the data into monthly groups, as done in reference [31]. This will help readers to better understand the developmental changes in muscle activity across different age groups.

In the previous study, we performed correlations of EMG activity during the whole period of SM recording (which also includes EMG activity during quasi-isometric contractions, very small movements, and very short bursts), while in this study we identified movement units from kinematic (video) recordings and specifically focused on the analysis of only movement-related EMG activity during the episodes of relatively large endpoint movements. This also explains some differences in the reported results. We specified it at the end of Introduction and in the Methods. Such reliable tracking of kinematics using video recordings and reliable reconstruction of endpoint motion could only be done in part of infants (n=25) (we now also added the data for 3 infants of 7 mo old so that we now 28 infants). Therefore, limitation of dividing the SM into 7 monthly groups (with the total number of registrations 30 [since two of 28 fullterm infants were recorded 2 times]) concerns the low number of children per age group (n=3-5). (we performed nevertheless this statistical test and still found a significant effect of age for TA-GL as in Fig. 6C, but for RF-BF it did not reach a significant level)

Therefore, to fix this limitation, as you also suggested, we now divided the infants into similar groups as for the analysis of passive movements, i.e. 0-3 mo (SM_Group 1) and 3-7 mo (SM_Group 2). We updated Table 2 and figures accordingly.

We now specified in the Methods:

“As for the analysis of spontaneous movements in fullterm infants, this was different in our previous study [51], since we did not identify the episodes of limb (endpoint) movements. In the present study, we focused our analysis on movement-related EMG characteristics (not including ‘isometric’ contractions or small movements) and, accordingly, we examined the data of the infants in which we reliably identified the specified episodes of limb movements (see below).”

and added to the Discussion (p19):

“In our earlier study [51], in which we looked at muscle-muscle correlations over the entire time of SM recording, we found that muscle activity correlations increased with age for fullterm infants (mostly for distant muscles). The current findings further demonstrate that this effect is seen in the antagonist muscles during the selected episodes of prominent limb movements (i.e., excluding 'isometric' contractions and minor movements).”

[4.2. Manifestation of muscle responses during the first year of life at page 17 of 21]

You seem to compare the number of spontaneous MU with the passive one determined by the experimenter. Please rewrite it as it was a little confusing.

As suggested, we modified this text to make it clearer:

“Since the infants were examined while awake and alert, it is first crucial to distinguish between muscle responses to passive movements and EMG signals during spontaneous movements. Indeed, some interference of StR and ShR with ongoing spontaneous activity cannot be ruled out. Nevertheless, it is unlikely that the muscle activity recorded during passive movements was random, i.e., unrelated to the passive movement, for the following reasons. First, in order to achieve the most relaxed state of the infant, typically we started recording passive movements when the infant was almost motionless (for ~5 to 10 s prior to registration), that is, when there were no evident spontaneous movements. Second, even though the experimenter determined the frequency (number) of passive movements (with flexion/extension movement duration ~1.5–2 s, Fig. 1B), the occurrence of StR and ShR within the total interval of passive movements recording was significantly higher than the occurrence of spontaneous movements. In particular, the occurrence of the episodes of SMs was about 6-8 MUs/min (Fig. 6A left panel), while the occurrence of StR and ShR was much more frequent, approximately every one or two movement cycles (see percent of movements, Fig. 2A), which roughly corresponds to ~20-40 muscle responses per minute. Third, and most importantly, these reactions to passive movements were linked to a particular phase of flexion/extension motion, and persisted for several cycles (Fig 1C, 3A-B), meaning that the onset of StR and ShR did not occur randomly throughout flexion/extension motion (Fig. 4B) but had a latency of ~0.5 s (Fig. 4A). Finally, cyclic flexion/extension movements in one joint could also evoke regular rhythmic muscle responses in distant joints in younger infants (Fig. 4). Therefore, we believe that the reported muscle reactions to passive joint motions accurately reflect how muscles respond to gradual changes in muscle length as indicators of muscular tone.” 

<Minor points>

I attached the PDF file with my minor comments.

Thank you for these comments as well! To ease the reviewer’s inspection, we provided the responses for each comment below:

***************************

Simple summary

disturbances to intrauterine growth

The "disturbance of intrauterine growth" recalls the term 'IUGR', which has different meanings of term "preterm" and must always be used separately.

As suggested, we replaced it with “shorter gestational periods

muscle responses during passive and spontaneous movements

As suggested, we replaced it with “muscle responses to passive movements and muscle activities during spontaneous movements”.

It is recommended that consistent terminology is used. "infant" seems to be a good choice here.

Done

Abstract

muscle power

 Muscle power and muscle tone are different (and they are separated in the main text) and should be rewritten or explained here as they are confusing

Thank you for this comment. Muscle power and muscle tone are indeed different, and we simply meant here (and in the main text) that some aspects of muscular development (e.g., muscle power development) may occur differently in preterm infants than in infants who are born at term. To avoid this confusion, we now omitted ‘muscle power development’ from the abstract and slightly changed the text:

“In preterm infants, some aspects of muscular development may occur differently than in infants born at term.”

And in the main text (last paragraph of the Introduction), we mentioned about it accordingly:

“In preterm infants, some aspects of muscular development, such as muscle power development, may follow a different time course compared with that of infants born at term”.

compared them to those in fullterm infants

The part of data might be from the authors previous study and should be clearly stated here

As suggested, we now mentioned in the abstract: “… and compared them to those reported in our previous study on fullterm infants”.

corrected-age months

I feel that it should be modified a little bit because it is not often heard in this way.

As suggested, we omitted “corrected-age” here.

Introduction

interactive (e.g., mother-child)

Note that [6-9] does not include such content, so please add references on this point.

We added it.

[10,11].

Papers that directly refer to such phenomena should be refered. Animal experiments or model studies are acceptable.

We added some references on animal experiments.

childhood

infancy

Done

disturbances to intrauterine growth. Such disturbances may be temporal or persistent

As noted in the Abstract, the meaning is different.

Here, it is better to simply say shorter gestational age/period, or reduced intrauterine experience, etc.

As suggested, we replaced it with “shorter gestational periods”.

tonic muscle reactions

Describe the definitions of "tonic muscle reaction/activitiy" (compared to phasic and reflex one), which do not seem to be adequately described in this paper.

As suggested, we now defined it in the previous paragraph:

“It is common to think of the way muscles react to slow changes in muscle length as signs of muscular tone [31–38]. Such reactions can be observed at rather slow velocities, and muscle activity may persist after the movement has stopped. Therefore, they are often referred to as tonic muscle reactions, or as being related to a redistribution of muscle tone across antagonist muscles [39–42], as opposed to reflex responses to brief perturbations.”

To our knowledge, tonic muscle reactions in human infants have not been systematically investigated. In a subgroup of participants, we also assessed spontaneous muscle activity during the episodes of relatively large limb motion amplitudes (comparable with those during evoked passive movements)

You have your own paper and you should change the wording.

I think that it is first time to compare them between preterm and full-term infant.

We replaced the sentence “To our knowledge, tonic … investigated” with:

“To our knowledge, the characteristics of tonic muscle reactions between preterm and fullterm newborns have not been compared. However, we have previously described the muscular responses to passive movements in infants who were born at term [25].”

Materials and Methods

It is worrying that the description of methods is too similar to previous studies.

We paraphrased some text.

Participants were 50 healthy fullterm infants

It should be clearly stated that the data are from previous study[23], if they are almost the same.

In that case, the reader also would be curious about why some of them were excluded.

As we responded above, now we presented the data of the same infants as in our previous study and updated the figures accordingly.

uneventful delivery and perinatal history

The latter also needs to be denied, so rewrite like following "without derivery event or oerinatal history".

Done

39.5±0.6 weeks

It was a little worrying that the number of subjects changed from the previous study[23] but the SD remained unchanged.

As we responded above, now we presented the data of the same infants as in our previous study.

clinically stable

Which timing? At birth, measurement or inclusion?

at measurement

preterm birth >25 weeks of gestational age

The terminology is wrong. You can remove "preterm" here.

Done

who had undergone alterations of the central nervous system

We did not know what they were referring to. Can you give specific examples?

We omitted this text (we meant preterm infants, who had abnormal neurodevelopmental outcome).

Child babies baby

infant

Done

spontaneous movements in these infants was different

Does this mean that the 40infant in [31] and the 28infant in this study are completely different?

Thank you for this comment. Indeed, it was likely not clear in our submitted ms. We now specified it (page 3):

“As for the analysis of spontaneous movements in fullterm infants, this was different in our previous study [51], since we did not identify the episodes of limb (endpoint) movements. In the present study, we focused our analysis on movement-related EMG characteristics (not including ‘isometric’ contractions or small movements) and, accordingly, we examined the data of the infants in which we reliably identified the specified episodes of limb movements (see below).”

and page 6:

“DeepLabCut could only successfully track and reconstruct the endpoint motion of a portion of the fullterm infants (n=25) that we had previously registered (n=40 in [51]). We have now included the data for 3 infants who were 7 months old, bringing the total to 28 fullterm infants.”

(Table 1)

The infant group in Table 1 should be almost the same as in the previous study [23], but only Group2 has a few fewer members and the results are different.

It is necessary to explain that this difference is not an arbitrary manipulation.

As we responded above, now we presented the data of the same infants as in our previous study.

within 6-7 months only [16,31],

We replace it with ‘5-7’ and added another reference.

0-2 months, 2-4 months and 4-7 months (Table 2)

Can you divide the spontaneous movement into monthly groups as cited in reference 31 or add that data to the supplement? Also, please group them by the same age range as the first experiment to allow for comparison with the muscle response to passive movement at the same age.

As we mentioned above, in this study we analyzed less fullterm infants during SM (n=28) than in our previous study (n=40). In the previous study, we performed correlations of EMG activity during the whole period of SM recording (which also includes EMG activity during quasi-isometric contractions, very small movements, and very short bursts), while in this study we identified movement units from kinematic (video) recordings and specifically focused on the analysis of only movement-related EMG activity during the episodes of relatively large endpoint movements. This also explains some differences in the reported results. We specified it at the end of Introduction and in the Methods. Therefore, limitation of dividing the SM into 7 monthly groups (with the total number of registrations 30 [since two of 28 fullterm infants were recorded 2 times]) concerns the low number of children per age group (n=3-5). (we performed nevertheless this statistical test and still found a significant effect of age for TA-GL as in Fig. 6C, but for RF-BF it did not reach a significant level) To fix this limitation, as you also suggested, we now divided the infants into similar groups as for the analysis of passive movements, i.e. 0-3 mo (SM_Group 1) and 3-7 mo (SM_Group 2). We updated Table 2 and figures accordingly.

ВВ

We recorded the acceleration signal in BB but we did not use it. We specified in the following section (2.4.1) that “Since the arm and shank segments (where we recorded the EMG and IMU signals) were essentially motionless during movements of the distal segments (forearm and foot, respectively, Fig. 1A), passive movements for the elbow and ankle joints were analyzed by two experimenters independently (with good correspondence) using video recordings.”. But to avoid a potential confusion, we now eliminated BB from this text.

First, the baseline level of muscle activity was assessed by calculating the mean EMG level of all signal fragments of at least 0.5 s in length, for which the average EMG amplitude was no more than 1.2 times larger than the minimum level throughout the trial (assuming that it represents or is close to the level of noise in the individual muscle recordings).

If the threshold of the onset of muscle activity is the mean of the measured interval, the threshold should be higher than the general (frequentlu used) threshold (3 SD at rest) and the sensitivity of the beginning of mscle activity would be lower. Additional explanation is needed why this definition was adopted.

Also, please add an explanation what you mean by 1.2 times the minimum value, as the meaning/biological meaning in an experimental system is unclear.

3SD at rest, as the threshold ‘frequently’ used for the onset of muscle responses, is likely reasonable to use when there is actual background EMG activity prior to the stimulus or movement, while in our case the calculated baseline level was close to or equal to the level of noise in the individual muscle recordings and therefore we adopted a different criterion. (in fact, the threshold we used - 2 times higher than the ‘baseline’ level - was greater than ‘3 SD at rest’)

Likely, our previous description was somewhat incomplete or unclear. Therefore, we now slightly modified this text and replaced “baseline level” with “the fragments of low level (baseline) activity”:

“First, we started by identifying the fragments of low-level (baseline) activity lasting at least 0.5 s, for which the average EMG amplitude was no more than 1.2 times greater than the minimum level throughout the trial (i.e., was close to or equal to the level of noise in the individual muscle recordings). Then, we …”

Also, in our previous study (Solopova et al 2019) we looked at how sensitive the outcomes were to the precise threshold value and we reported that: “We used non-normalized (in µV) EMG data to determine StR and ShR, However, even when we considerably increased the threshold for detecting the periods of muscle activity (three times exceeding the baseline level instead of two times, see Methods), the percentage of detected muscle responses decreased (by 25%), however, the effect of age … remained the same for most muscles.”

We believe that our adopted criterion (“2 times higher than the baseline level”) is relevant and preferable to using 3 SD at rest (since the “biological meaning” of ‘3 SD’ around the level of noise might be problematic).

~10 cm

Is the intention to say 'more than' when matching 15% or more? This can make it look like 'less than 10 cm'.

We replaced it with ‘more than 8 cm’ (the value ‘in cm’ depends on the age [for youngest infants, it’s about 8 cm, for older infants, it’s >10 cm]. We normalized the threshold by the body height.)

∑большая формула для индекса коактивации

Explanation of N is needed. Is it "4" here?

Done (N is the number of temporal points digitized in the MU)

EMGH and EMGL

the highest and the lowest activity betwee

subscript

Done

Results

passive rotations

The term "rotation" evokes a specific joint movement, so use a different term (i.e. passive joint movements) to avoid misunderstandings in whole manuscript.

We replaced it as suggested.

the calculation of StR and ShR durations

It looks different from the definition in [23].

In [23], the Duration of StR is only for the duration of the flexion, but the way this diagram is presented, it appears to be independent of the duration of the flexion, and in some cases it is likely to exceed 100%.

The duration was estimated in the same way as in our previous study. Indeed, in some cases, muscle activity continued after a change in the direction (it can also be observed in adults, when ‘tonic’ muscle activity during slow muscle lengthening or shortening may continue after the movement has stopped or changed in the direction), and we specifically illustrated some examples of muscle responses in Fig. 1C, as well as in Fig. 3A.B. We specified in the Methods: “. If muscle activity persisted following a change in direction of movement (for example, from flexion to extension), the duration was only calculated from the time it began to the end of this phase.

threshold

Please state clearly that the dotted line is the threshold.

Done

example of identified MUs.

Please state clearly which endpoint this example is.

Done

Major types of reactions to passive flexion/extension movements were described in Solopova et al. [23]

Readers do not always read previous studies, so they should be described as a result of this study and not rely on statements from previous studies. Also, please add the discussion about the differences of the data in this experiment with the data presented in reference [23] in discussion section, particularly in terms of full-term? Although this is the same experiment, there are some differences, notably the significant variation in incidence at 9-12 months.

We modified the text. As we responded above, we now presented the data and corrected figures for the same number of fullterm infants as in our previous study.

Briefly, they included mainly ShR or StR in one muscle with no concurrent activity in the antagonist (Fig. 1C, left panel), or more complex responses including the presence of both StR and ShR in antagonist muscles (Fig. 1C, right panel). Since most of our recorded muscles were bi-articular, activity in the same muscle could be evoked during movements in different joints. For instance, ShR in RF was observed during flexion in the hip joint and extension in the knee joint, while ShR in BF was observed during hip extension and knee flexion.

You are describing major types here, but please describe them more clearly as it is difficult to understand their definitions.

We modified and shortened the text to make it clearer:

“They could include mainly ShR or StR in one muscle with no concurrent activity in the antagonist (Fig. 1C, left panel), or simultaneous StR and ShR in antagonist muscles (Fig. 1C, right panel). Since most of our recorded muscles were bi-articular, ShR in RF was observed during flexion in the hip joint and extension in the knee joint, while ShR in BF was observed during hip extension and knee flexion (Fig. 2A).”

(Fig 2A)

Does this mean that all ages and muscles were compared together? If so, please specify.

We added: “when comparing elbow joint movements to those of the hip, knee and ankle joints separately”.

their occurrence significantly decreased with age in most muscles in both preterm and
fullterm infants (p<0.03, post hoc Mann-Whitney U test)

If you compare the 9-12 group with the rest of the group, it would look clear. However, because some muscles seem to increase in the 3-6 group, it is not clear.

To make clear, draw a scatter plot for each muscle, divided into STR and SHR, with age on the horizontal axis and % on the vertical axis, if Supplementary is acceptable.

We agree that the effect of age for StR and ShR in individual muscles was not clear in this figure. Also, as we mentioned above, we apologize for the error in ‘scaling’ of the data for group 4 which indeed overemphasized the effect of age: we now corrected this error in new Fig 2A (statistics did not change, we only corrected the bar plots). Therefore, as the reviewer suggested, in order to better highlight the effect of age in responses of specific muscles, we now adopted the same format of Fig 2A as in our earlier publication (significant differences for the effect of age are indicated by horizontal lines).

in the 3-6 mo age group

and BT BB in 6-9 mo?

For TB and BB, the differences between the groups (preterm-fullterm) did not reach a significant level.

(ShR and StR in RF and BF leg muscles)

please add the value or bar plot in figure.

Done

were detected both when all infants' passive movements were pooled together (Fig. 2A), and

Is this something that has already been mentioned? If so, please delete it.

We deleted it.

when the percentage of infants who demonstrated ShR or StR in at least one or more movement cycles was plotted (Fig. 2B).

The point here should be that the proportion is higher in the preterm group, so please specify this. Also, I think a chi-square test is used for comparison or anything.

As suggested, we specified it and added a chi-square test:

“The above-mentioned differences between preterm and fullterm infants (group of 3-6 mo, Fig. 2A) were also detected when the percentage of infants who demonstrated ShR or StR in at least one or more movement cycles was plotted (Fig. 2B): both for ShR and StR, the proportion of such trials was higher in the preterm group (p<0.001 for ShR and p<0.002 for StR, chi-square test).”

Figure 2A

The color of the bar is not consistent for full-term. Also, I recommend using white for full-term and gray for preterm.

We now use consistent colors in all figures.

Also, there were no significantly difference in latencies of muscle reactions between preterm and fullterm infants (p>0.4 Mann-Whitney U test)

Data values should be provided in the figures or text.

We added new Fig. 3 with the latencies of StR and ShR for preterm and fullterm infants.

although the latencies of StR and ShR were similar for all joints.

I don't understand what the second half is intended to be. How is it similar compared to what? Delete the second half if there is no intention, as it is confusing as it states that there is a difference, at least immediately before.

As suggested, we deleted the second half of the sentence. Instead we added a new figure for the latencies for StR and ShR for each joint (Fig. 3).

~28% and ~40% of the flexion/extension phase)

As it is the experimenter who controls the speed of the passive motion, there seems to be no meaning in using percentage to describe latensy.

If there is, explain the intention, if not, delete it.

As suggested we deleted it.

Figure 3

(A) – right panel:

Left?

Thanks for pointing to this typo. We corrected it.

pie charts

Please provide numbers.

Done.

3.3. Selected movement episodes (MUs) of spontaneous activity

We used the criteria for selecting movement episodes during spontaneous activity in such a way that they reflected only the moments of pronounced motion of limb endpoints (wrist and foot), when the infants moved their limbs actively with sufficient amplitude and speed (Fig. 1E, see Methods).

I think this is already described in Method section, which can be omitted with just indicating 'see Methods'.

As suggested, we deleted this text (indicating ‘see Methods’).

Figure 4.

Note the presence of prominent though variable SMs in all four limbs in all infants

It was not clear to me what kind of change you wanted to show here, so please describe what you want to show specifically.

We omitted this sentence from the figure legend (we noted about it in the main text).

~6-9 MUs per minute

per joint?

No. As we described in the methods, we identified MUs based on leg and arm endpoint movements (we did not record 3D movements in different joints).

Figure 5.

separated

Is the arrow and area in the middle pointing in the wrong place?

The arrow and the area are correct. (according to our definition of separated MUs. The right leg was chosen as a referent leg)

(D)

This should be clearly stated about the corrected age of this data.

You should also include data of the fullterm infant group for the same age.

In these subplots, the x-axis refers to the gestational age, not to the age when SMs were recorded, so that we cannot add here the fullterms’ data (while panels A-C include the fullterms’ data).

Nevertheless, it is worth stressing that a great amount of EMG activity was associated with identified MUs (Fig. 4B,C), as well as video camera captured a great part of typical SMs in all infants (Fig. 4A).

By the looks of the figures, it is difficult to understand the association between EMG and MU, is there a better example? Or, please add more detailed explanations.

Also, please rewrite the second half of the diagram as it is difficult to understand the intention.

To better associate MUs with EMG in these examples (and that some muscle activity could also be observed in the intervals between MUs), we added patches of shaded green area to panels B and C. We also omitted the second half of this sentence (since we previously already mentioned that we identified MUs in the plane of video camera).

However, the analysis revealed differences in the effect of age

What does it mean?

We deleted this sentence and slightly shortened and modified the text in this paragraph to make it clearer.

connections

correlation?

We replaced it

Discussion

spontaneous (Fig. 4-5)

EMG during spontaneous movements do not only reflect muscle responses but also spontaneous/actve EMG.

Then, you should clearly distinguish responce to passive movemnts from EMG during spontaneous movements .

As suggested, we now expanded the part of the Discussion related to the differentiation between muscle responses to passive movements and spontaneous activity (p17-18):

“Since the infants were examined while awake and alert, it is first crucial to distinguish between muscle responses to passive movements and EMG signals during spontaneous movements. Indeed, some interference of StR and ShR with ongoing spontaneous activity cannot be ruled out. Nevertheless, it is unlikely that the muscle activity recorded during passive movements was random, i.e., unrelated to the passive movement, for the following reasons. First, in order to achieve the most relaxed state of the infant, typically we started recording passive movements when the infant was almost motionless (for ~5 to 10 s prior to registration), that is, when there were no evident spontaneous movements. Second, even though the experimenter determined the frequency (number) of passive movements (with flexion/extension movement duration ~1.5–2 s, Fig. 1B), the occurrence of StR and ShR within the total interval of passive movements recording was significantly higher than the occurrence of spontaneous movements. In particular, the occurrence of the episodes of SMs was about 6-8 MUs/min (Fig. 6A left panel), while the occurrence of StR and ShR was much more frequent, approximately every one or two movement cycles (see percent of movements, Fig. 2A), which roughly corresponds to ~20-40 muscle responses per minute. Third, and most importantly, these reactions to passive movements were linked to a particular phase of flexion/extension motion, and persisted for several cycles (Fig 1C, 3A-B), meaning that the onset of StR and ShR did not occur randomly throughout flexion/extension motion (Fig. 4B) but had a latency of ~0.5 s (Fig. 4A). Finally, cyclic flexion/extension movements in one joint could also evoke regular rhythmic muscle responses in distant joints in younger infants (Fig. 4). Therefore, we believe that the reported muscle reactions to passive joint motions accurately reflect how muscles respond to gradual changes in muscle length as indicators of muscular tone.”

[43]

You must refer articles on commonly used assessment methods in neonatal care.

For example, Dubowitz neurologic examination, Hammersmith Infant Neurological Examination, Modified Ashworth or Modified Tardieu Scales, and so on. 

We added a reference to these examinations.

the correlations of EMG activities of antagonist muscles were on average rather low.

With which data can you suggest this?

We added:

“Also, even though some amount of crosstalk can be present in the surface EMG recordings, the correlations of EMG activities of antagonist muscles were rather low on average (e.g., see Fig. 6C), and we often observed separate StR and ShR bursts in the antagonist muscles (Fig. 1C,4B). It is also worth noting that crosstalk could not account for frequent rhythmic responses in distant muscles of the ipsilateral (Fig. 4A) and contralateral limbs (Fig. 4B).”

obvious in both X and Y directions

What is the point of this?

We deleted this sentence.

Fir\

Done

almost in every movement cycle in some muscles or every 2-3 cycles in other muscles (see percent of movements, Fig. 2A), which roughly corresponds to ~20-40 muscle responses to passive movements per minute (given the duration of movement ~1.5-2 s, Fig. 1B)

Are you comparing the number of spontaneously occurring movements (MU) with the number of passive movements determined by the experimenter? If so, the comparison seems meaningless. If not, please rewrite it as it was a little confusing.

See our response above. (as suggested, we modified this text)

Interestingly, cyclic flexion/extension movements in one joint could also evoke rhythmic muscle responses in distant joints in younger babies (Fig. 3)

This may be influenced by the excitability of the pattern generator. It would be good to discuss this, citing papers on locomotor primitives.

Thank you for this suggestion. We added:

“Interestingly, in younger infants, flexion/extension movements in one joint could also cause rhythmic muscle responses in other ipsilateral or bilateral muscles (Fig. 4). This may be influenced by the excitability of the pattern generator circuitry, given the significant functional reorganization of the spinal locomotor output in early infancy [23,30] and the fact that its higher responsiveness is associated with greater H-reflexes [68], which are enhanced in younger infants [66,67]. The so-called “irradiation” of responses to mechanical stimulation to distant muscles in infants [26,27] may also contribute to the appearance of rhythmic responses in distant muscles (Fig. 4).”

higher excitability of spinal or supraspinal circuits

Please describe and refer to research that provides specific evidence for 'higher excitability of neural circuit'.

We added some examples and references:

“Our data support other age-dependent changes in excitability of spinal or supraspinal circuits, such as a reduction in the monosynaptic H-reflex responses [66,67] or in the incidence of response to mechanical stimulation [43] with increasing age.”

learning the functionally appropriate muscle tone during the first months of life,

Please describe and refer to research that provides specific evidence for 'functional muscle tone learning'.

We added (p19):

“The reason why the circuits are more excitable early on may be related to higher excitability of spinal or supraspinal circuits ([3,66,67]) and learning the functionally appropriate muscle tone (such as maintaining the stationary limb and body postures, developing a dynamic postural frame, antigravity posture control) during the first months of life, representing an important phase of maturation of the central nervous system and the processing of sensory information. Variability in the occurrence of ShR vs. StR and their characteristics (Figs. 2-3) may also be functional and serve to learn to set stable feedback gains [27] and to develop adaptive motor behavior [77] in early infancy.”

in the first months 

0-2? or earlier age

We slightly modified this sentence to specify the details:

“However, the preterm infants showed higher rates of muscle responses at 3-6 mo corrected age, and lacked age-related increases in antagonist activity correlations during SM (Fig. 2A-B and 6C, respectively).”

similar to

You did not test the similarity, then you have to describe like "not different".

Done

The alterations

What kind of alterations are being referred to here? Also, since this phrase may be confused with the term 'developmental change', it may be better to rephrase it. Atypical response?

We replaced it with “atypical responses” as suggested.

Reviewer 2 Report

The manuscript by Dolinskaya and colleagues examined muscle activity characteristics resulting from passive and active movements in infants under 1 year of age. The results indicated a decrease in electromyographic responses with age in passive movements, and an increased correlation of agonist-antagonist leg muscle pairs during spontaneous movements. One of the strengths of the manuscript is the objective identification of a spontaneous movement in the infants using an algorithm. The manuscript presents some interesting results, and I am in support of publication. I have the following comments.

My main comment regarding the manuscript is the labeling of muscle “stretching” and “shortening”, since shortening of an agonist muscle would induce stretching of the antagonist muscle. It seems that this these ideas are not completely separated in the analysis and discussion. I think this may be important since it seems to me that these responses are likely related to muscle spindle activity. I wonder if there is any value in addressing this aspect in the manuscript. On this note, I also wonder if the authors are able to comment on the excitability of Ia and II afferents from the muscle spindle? The methods state that the passive movements are performed “slowly”, so would this be more reflective of the static components of the muscle spindle?

Minor Comments:

Abstract:

-       Clarify what is meant by “distant” muscles. Maybe another term is more appropriate?

      Introduction:

-       Same as above: “agonist and distant muscles”. Is there any other terms that could clarify what is meant by this as “distant” is somewhat ambiguous.

-       Clarify: “Such disturbances may be temporal or persistent.”

Methods:

-          Clarify “corrected age”. How was this calculated?

-          Data analysis: “interference” EMG were rectified and smoothed??

-          Please provide more information on the correlation analysis. Specifically, what exactly was correlated here?

Author Response

The manuscript by Dolinskaya and colleagues examined muscle activity characteristics resulting from passive and active movements in infants under 1 year of age. The results indicated a decrease in electromyographic responses with age in passive movements, and an increased correlation of agonist-antagonist leg muscle pairs during spontaneous movements. One of the strengths of the manuscript is the objective identification of a spontaneous movement in the infants using an algorithm. The manuscript presents some interesting results, and I am in support of publication. I have the following comments.

We thank the reviewer for his/her evaluation of our study, and suggestions served to improve the manuscript. Changes in the text are marked in red.

My main comment regarding the manuscript is the labeling of muscle “stretching” and “shortening”, since shortening of an agonist muscle would induce stretching of the antagonist muscle. It seems that this these ideas are not completely separated in the analysis and discussion. I think this may be important since it seems to me that these responses are likely related to muscle spindle activity. I wonder if there is any value in addressing this aspect in the manuscript. On this note, I also wonder if the authors are able to comment on the excitability of Ia and II afferents from the muscle spindle? The methods state that the passive movements are performed “slowly”, so would this be more reflective of the static components of the muscle spindle?

Indeed, during the same flexion or extension movement one muscle is shortening and its antagonist is lengthening (and vice versa). We now tried to make it clearer though the manuscript and also discussed the coexistence of ShR and StR in the Discussion (p19). We also mentioned in the Introduction that both reactions (StR and ShR) have been discussed in the literature in the context of a redistribution of muscle tone among antagonists (p2):

“It is common to think of the way muscles react to slow changes in muscle length as signs of muscular tone [31–38]. Such reactions can be observed at rather slow velocities, and muscle activity may persist after the movement has stopped. Therefore, they are often referred to as tonic muscle reactions, or as being related to a redistribution of muscle tone across antagonist muscles [39–42], as opposed to reflex responses to brief perturbations. Muscular responses in infants to mechanical perturbations (e.g., responses to tendon taps [27]) or nerve stimulation (e.g., cutaneous flexion reflex [43]) are typically associated with ‘reflex’ reactions and have relatively short latencies, while muscular responses to slow changes in muscle length may differ in their functional role and underlying mechanisms. It has also been argued that maintaining a limb posture following movement involves distinct neural circuits [44,45]. Muscle tone can be manifested in both resistive and ‘compliant’ behavior, related to movement as a state is related to an action [46]. Muscle responses to passive stretching (StR) and shortening (ShR) may reflect resistive and compliant behavior. Compliant behavior has an important functional significance. Forster [47] was likely the first who considered the functional role of ShR, viewing it as a muscle-length adaptation reflex. Therefore, analyzing both types of responses (StR and ShR) can provide valuable insights into how muscle tone manifests during typical and non-typical motor development.”

As suggested by the reviewer, we also commented on the participation of different proprioceptive inputs in the Discussion (p18):

“The way muscles respond to slow changes in muscle length, StR or ShR, is frequently addressed as expressions of muscle tone and its redistribution among antagonists [39–42]. While the responses to lengthening are most likely associated with the excitability of II afferents from the muscle spindles, the neural substrates of the ShR are not fully understood. The ShR can be induced at relatively low velocities of imposed joint rotations, and the role of Golgi tendon organs [70], group II muscle afferents [71], and joint receptors [34] in its generation have been previously discussed in the context of its functional role as an adaptation reflex of muscle to its length [47]. It has been also noted that ShR is accompanied by a discharge of primary endings of muscle spindles due to α-γ coactivation [72]. Since ShR already belongs to an innate repertoire of compliant motor behavior (observed in infants as young as 0-3 mo, Fig. 2A), when cortical control is still immature and limited [73,74], it is reasonable to suggest that it mainly depends on subcortical mechanisms and sensory inputs.”

Minor Comments:

Abstract:

-       Clarify what is meant by “distant” muscles. Maybe another term is more appropriate?

As suggested, we briefly explained and replaced here “distant muscles” with “muscles not being primarily stretched/shortened“.

      Introduction:

-       Same as above: “agonist and distant muscles”. Is there any other terms that could clarify what is meant by this as “distant” is somewhat ambiguous.

We replaced it here with “other” although we use this term later in the manuscript (we hope it is clearer now) since the term “distant muscles” has been previously used in the context of irradiation of sensory-evoked responses to distant muscles in infants and we cited such references.

-       Clarify: “Such disturbances may be temporal or persistent.”

We added (p2):

“Such disturbances may be transient (only noticeable, for instance, during the first few months of life) or persistent [50].”

Methods:

-          Clarify “corrected age”. How was this calculated?

As the infant’s age corrected for degree of prematurity (i.e., chronological age reduced by the number of weeks born before 40 weeks of gestation)

-          Data analysis: “interference” EMG were rectified and smoothed??

We slightly modified and shortened this sentence: “The EMG signals were rectified and then smoothed by a sliding 20-ms window …”

-         Please provide more information on the correlation analysis. Specifically, what exactly was correlated here?

We added (p.8):  

“The coefficient of linear correlation (r) of EMG activity was analyzed for selected pairs of muscles (RF-BF, TA-GL and BB-TB) during the identified episodes of spontaneous movements (during MUs), and averaged across MUs to illustrate age-related differences (assessed using non-parametric statistics like the Kruskal-Wallis test with Holm-Bonferroni correction).”

Round 2

Reviewer 1 Report

The authors have been very responsive to the comments raised in the review and thorough in their response. I have no further comment.